# Global greenhouse gas emissions from residential and commercial building materials and mitigation strategies to 2060

Xiaoyang Zhong [1]✉, Mingming Hu[1,2], Sebastiaan Deetman [1,3], Bernhard Steubing [1], Hai Xiang Lin[1,4], Glenn Aguilar Hernandez [1], Carina Harpprecht [1,5], Chunbo Zhang [1], Arnold Tukker[1,6] & Paul Behrens[1,7]✉

Building stock growth around the world drives extensive material consumption and environmental impacts. Future impacts will be dependent on the level and rate of socioeconomic development, along with material use and supply strategies. Here we evaluate material-related greenhouse gas (GHG) emissions for residential and commercial buildings along with their reduction potentials in 26 global regions by 2060. For a middle-of-the-road baseline scenario, building material-related emissions see an increase of 3.5 to 4.6 Gt CO2eq yr-1 between 2020–2060. Low- and lower-middle-income regions see rapid emission increase from 750 Mt (22% globally) in 2020 and 2.4 Gt (51%) in 2060, while higher-income regions shrink in both absolute and relative terms. Implementing several material efficiency strategies together in a High Efficiency (HE) scenario could almost half the baseline emissions. Yet, even in this scenario, the building material sector would require double its current proportional share of emissions to meet a 1.5 °C-compatible target.

[1] Institute of Environmental Sciences (CML), Leiden University, 2333 CC Leiden, The Netherlands. [2] School of Management Science and Real Estate, Chongqing University, Chongqing 40045, China. [3] Copernicus Institute for Sustainable Development, Utrecht University, 3584 CB Utrecht, The Netherlands. [4] Delft Institute of Applied Mathematics, Delft University of Technology, 2628 CD Delft, The Netherlands. [5] German Aerospace Center (DLR), Institute of Networked Energy Systems, Curiestreet 4, 70563 Stuttgart, Germany. [6] Netherlands Organization for Applied Scientific Research TNO, 2595 DA The Hague, The Netherlands. [7] Leiden University College The Hague, Leiden University, 2595 DG The Hague, The Netherlands.
✉email: x.zhong@cml.leidenuniv.nl; p.a.behrens@cml.leidenuniv.nl

Housing is one of the most immediate basic human needs, along with food and clothing[1]. The provision of residential and commercial buildings is responsible for over one-third of energy use and energy-related GHG emissions globally[2]. There are two main ways to mitigate building-related emissions: (1) decarbonize/reduce the energy needed for in-use buildings and (2) decarbonize/reduce the production of materials and energy in construction. Environmental policies have traditionally focused on improving energy efficiency and renewable energies in the use phase while neglecting material efficiency in construction[3,4]. A policy approach that focuses only on in-use emissions may miss important opportunities in construction[5,6]. Indeed, there may also be important tradeoffs between pre-use and in-use emissions whereby highly energy-efficient buildings may require more materials in construction[7–9]. In 2018, the manufacturing of building materials alone accounted for 11% of global energy- and process-related GHG emissions[2], as a result of consuming over half of global concrete and brick[10], some 40% steel[11], and a large number of other metals and nonmetallic minerals[12].

Global trends indicate a rapid increase in demand for new buildings in the coming decades. This is mainly driven by growing populations and increasing wealth around the world (especially in Asian and African regions[2,13]), but also due to a demand for housing upgrades in highly urbanized areas[14]. As such, large amounts of materials are needed. Building technology has advanced substantially over the past decades. For example, buildings can be built with lower environmental impacts (such as using wood[15] or less metal for the same structural properties[16]), designed for a longer lifespan[17], or for a higher post-consumer recycling rate[18]. However, despite these technological advances, less-efficient building practices are still being widely used, especially in regions that will see most of this demand[19,20]. These trends pose a critical challenge in reducing GHG emissions from building materials and meeting global climate targets.

Research on the environmental impacts of building materials and mitigation strategies has gained momentum only in the past decade. Studies have either focused on residential building materials in a single country[17,21–23] or represent a certain material type at one time[24–26]. Further, calculating emissions requires consistent scenarios of both materials demand and process emissions intensities[6], whereas most studies address just one of these aspects[27,28]. A recent study[29] assessed the climate impacts of materials efficiency strategies on residential buildings in 9 large economies. Though valuable, this study omitted most emerging African and Asian regions (which represent much of the increasing housing demand in the future[2,13]) as well as the global non-residential buildings.

Here we develop a global building material emission model that integrates a dynamic material assessment model for estimating future building materials demand, and a prospective life cycle assessment (LCA) model to estimate emissions from materials production. We include 7 materials in 4 residential buildings types and 4 commercial building types across 26 world regions (see Methods). We investigate the development of global GHG emissions of residential and commercial building material production. We investigate the impacts of major material efficiency strategies and the implications of these strategies for meeting climate targets (Methods). We find a continuous increase in building material-related GHG emissions on a global level and dramatically different emission trends across world regions. We observe significant emission reduction and material loop closing potentials in the considered material efficiency strategies. We outline important mitigation opportunities and challenges associated with building materials for achieving global climate targets.

## Results

**Scenario narratives.** We base our investigation on outputs from IMAGE[30,31], a globally integrated assessment model, and the ecoinvent[32] life cycle inventory database. Different shared socio-economic pathways (SSPs)[33] are modeled in IMAGE reflecting possible future developments of socioeconomic parameters. We select the "middle-of-the-road" SSP2 pathway[34] which expects a moderate population and GDP growth. We use the socioeconomic[30,31] and energy transition scenarios[35] under IMAGE-SSP2 as inputs for our dynamic building materials model and prospective LCA, respectively. We explore two scenarios for the development of material requirements and emissions to 2060: a Baseline scenario, given by the SSP2-baseline parameters from IMAGE, and a High Efficiency scenario, assuming full implementation of several important materials efficiency strategies drawn from the literature (see Table 1). The time period from now to 2060 is characterized by population rise with income converging across economies[30,33], which have dramatic impacts on building construction and material demands. It also gives the industry sufficient time to develop and scale-up technologies for a sustainable transition[36]. The literature supporting the feasibility of these strategies often provides a target by 2050, not 2060. In such cases, we extrapolate these targets to 2060. Please see Methods the Supplementary Information for full details on the model, data, and scenarios.

**Baseline emissions.** The Baseline scenario sees a continuous increase in building-material-related GHG emissions at a global average of 0.7% yr$^{-1}$ (from 3.5 to 4.6 Gt CO$_2$eq yr$^{-1}$) between 2020 and 2060. This trend varies significantly across income groups (see Fig. 1a, b). The low- and lower-middle-income group sees the largest increase from 750 Mt (22%) in 2020 to 2.4 Gt (51%) in 2060 (see Fig. 1b), mainly due to a surge in population and economic development. For example, India, the Rest of South Asia, and Africa (excluding South Africa) will more than double their material-related emissions from 2020 to 2060. By comparison, the high-income group sees a slight decline in absolute terms and a sharp fall as a proportion

---

**Table 1 Mitigation strategies for reducing emissions from materials required for buildings construction.**

| Strategies | Description |
|---|---|
| M1—More intensive use | 20% lower area per person compared to 2050 baseline[29] |
| M2—Lifetime extension | Up to 90% lifetime extension (depending on the region and average lifetime) by 2050[29] |
| M3—Lightweight design | 19% reduction in aluminum and steel, 10% in concrete by 2050[6,16,29] |
| M4—Material substitution | 10% more timber buildings by 2050[29,37] |
| M5—More recovery | Maximum recycling and reuse rates estimated by 2050(recycling: 90% steel[38], 95% aluminum[26], 93% copper[39]; reuse: 15% steel and concrete[6,29]) |
| M6—Energy transition | An energy transition consistent with the SSP2-RCP2.6[35] |
| M7—Production efficiency increase | Efficiency increases of material production via manufacturing improvements and process-switching (for example switching from hydrometallurgy to pyrometallurgy processes for copper production)[28,40–42] |

Strategies are drawn from the literature as feasible targets (see the second column for specific references). Please see the Supplementary Information for further information.

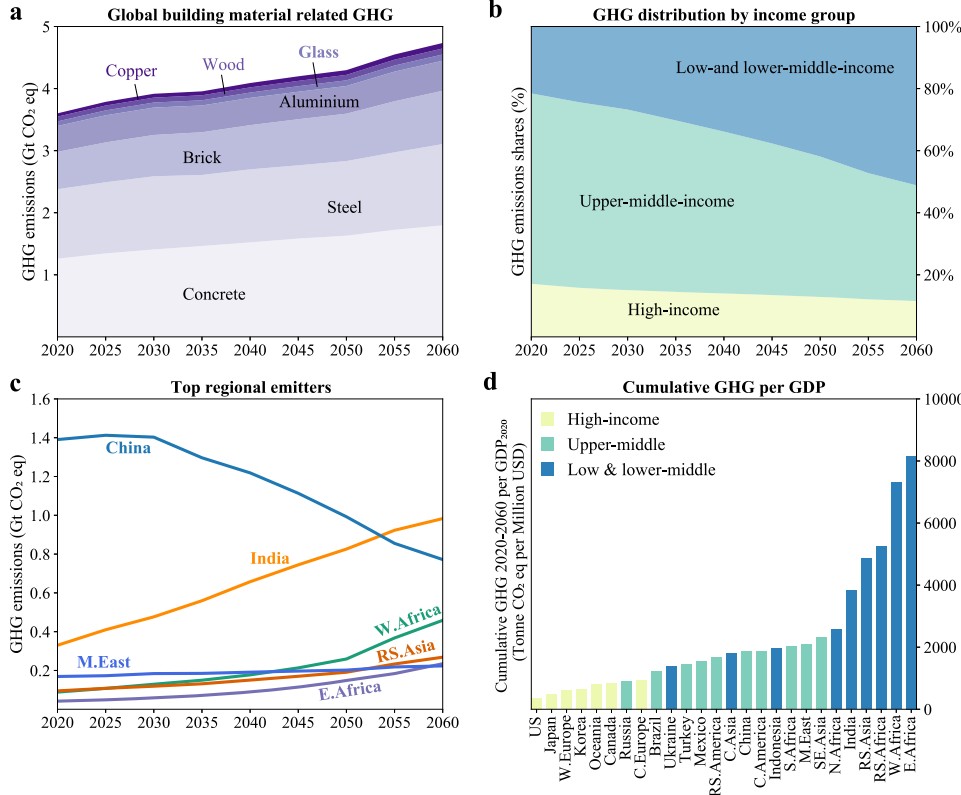

**Fig. 1 Greenhouse gas (GHG) emissions from building materials use for global regions in the Baseline scenario. a** Development of global GHG emissions for seven materials during 2020–2060. **b** Percentage evolution of GHG emissions for three income groups during 2020–2060. **c** Development of emissions in the top 6 emitting regions (by 2060), occupying over 60% of the total, during 2020–2060. **d** Expected cumulative GHG emissions over 2020–2060 relative to present GDP (2020 value from the IMAGE integrated assessment model, at purchasing power parity) for 26 global regions.

of global emissions, from 595 Mt (17%) in 2020 and 530 Mt (12%) in 2060. A similar trend is seen in the upper-middle-income group (Fig. 1c). Figure 1d shows the regional comparison of cumulative material-related GHG relative to GDP, highlighting contrasting economic challenges for the adoption of mitigation strategies. In general, high-income regions (such as the US, Japan, and Western Europe) will see relatively lower emissions and, therefore, have higher affordability of deep decarbonization.

The China region and India remain the top two emitters for the period 2020–2060, with India becoming the largest emitter by 2053 (Fig. 1c). The top 6 regional emitters in 2060 will all be in Asia or Africa (Fig. 1c). Overall, Asian regions see the majority (over 65%) of cumulative building material emissions over 2020–2060, followed by Africa at slightly over 10%. For material types, steel and concrete remain the largest emission sources at around two-thirds of the total, followed by brick (18%) and aluminum (8%) (Fig. 1a). The share of metal-related emissions sees a slight decrease from 43 to 39% over the period 2020–2060 likely due to an increase in secondary metals production.

**Strategies for emissions mitigation.** The mitigation potential of material efficiency strategies depends on the in-use building stock, construction practices, and the future techno socio-economic development in different regions. Figure 2 shows the reduction potential for each strategy at their High Efficiency levels during 2020–2060 (in comparison with the Baseline values and when each strategy is adopted independently of each other). In general, the reduction potential decreases from the top layer (building demand) down to the middle layer (material demand) and the bottom layer (material supply). That is, in terms of the

feasible interventions drawn from the literature, housing demand reduction has a higher potential for reducing impacts than improving material intensity, which in turn has a higher potential than increasing efficiency in the material supply.

Globally, more intensive use represents the largest emission reduction potential of 56.8 Gt $CO_2$eq as it simultaneously avoids a percentage of all materials. As a consumption-oriented strategy, more intensive use of the building stock represents the possibility to decouple the growth of buildings demands from economic development[20,44]. It does not necessarily lead to lower wellbeing and can be achieved by e.g., lower vacancy rates[45,46], more shared offices[47], and telecommuting[48]. As such, this strategy is heavily dependent on lifestyle and behavior transitions[20]. This potential is especially large in rapidly urbanizing regions such as China and highly urbanized regions like Western Europe, which will see shrinking populations and an opportunity to increase housing intensity[45,49].

Lifetime extension yields lower demands for new construction and emission reductions of 6.6 Gt globally. The opportunities for lifetime extension vary depending on the region. For example, although some older buildings can have their lifetimes extended in regions where the services life is very short (such as China, Japan, and Southeastern Asia), frequent demolition is often not due to construction quality but because of evolving urban planning and land policies[50–52]. Longer-lived buildings built today will only bring significant environmental returns decades later and only if planners ensure that the urban form is sustainable over the longer term. Poor urban planning can result in the lock-in of poor, unsustainable urban environments which would require demolition and reorganization in the future.

Light-weighting gives potential cumulative reductions of 14.1 Gt $CO_2$eq. This may be achieved by large-scale adoption of emerging

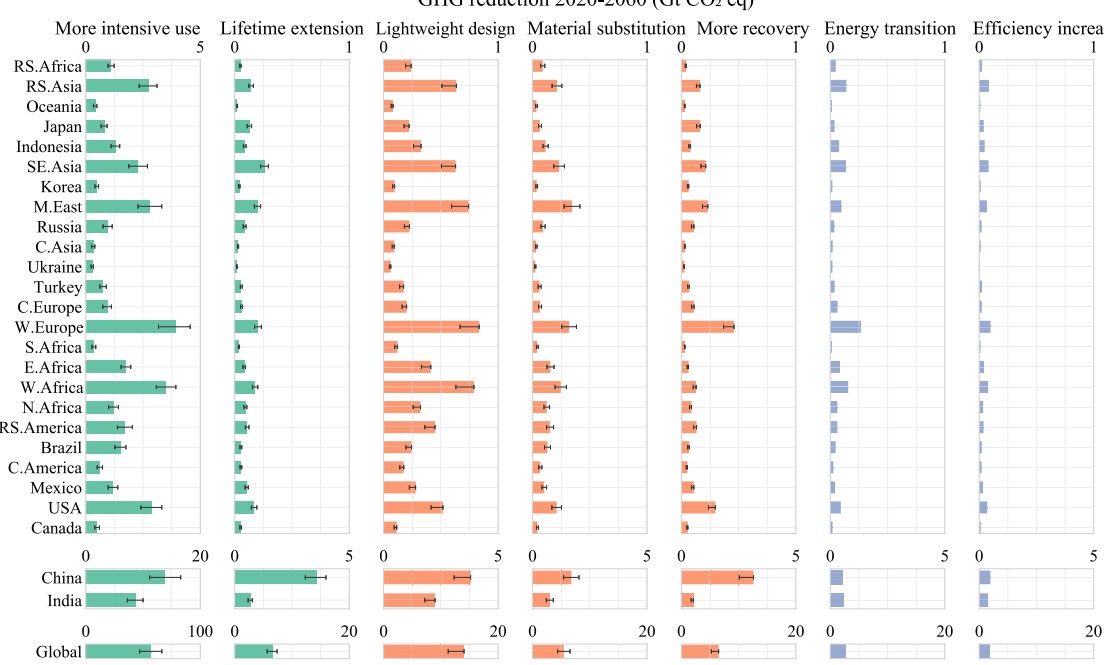

**Fig. 2 Greenhouse gas (GHG) emission mitigation potential during 2020–2060 by different material efficiency strategies.** The three colors left to right represent the three layers in the modeling framework: building demand, material demand, and material supply (see Supplementary Fig. 1). These three approaches correspond approximately to the general "avoid–shift–improve" emission mitigation framework[43]. The whiskers represent the sensitivity intervals of GHG in the High Efficiency (HE) scenario (given by 20 percentage point variations for each strategy; see the Supplementary Information for further details). Note that the scales for Global, the China region, and India differ from other regions, and the scale for 'more intensive use' differs from other strategies.

technologies including novel structural design[53], typology optimization[54], additive construction (such as 3D printing)[55], and the use of high strength steel and aluminum[5]. Some adjustment of building regulations is likely essential for such light-weighting transitions. Depending on the technologies and level of adoption there may be larger opportunities for light-weighting than those adopted in Table 1, e.g., 20% or more concrete reduction[29,56]. The current cost barriers to this implementation may reduce over time through deployment-led learning. Increasing the use of timber in buildings would result in GHG emission reduction of 5.5 Gt CO2eq (due to the lower emission intensity of timber production) and provide long-term carbon storage[37,57]. In a similar manner, secondary production of metals significantly reduces energy use and emissions, avoiding mining and early manufacturing emissions[28]. As post-use scraps become increasingly available, higher recycling and reuse play an increasingly important role in mitigation, with a cumulative potential of 6.5 Gt GHG over 2020–2060 (Fig. 2). To approach the maximum recycling potential, rapid up-front industrial investment is needed to develop both new technologies and supporting infrastructure[26,58].

In the material production stage, the energy transition (to decarbonize energy used in the background LCA system) and efficiency improvements (to reduce energy in the foreground LCA system) have the combined potential for reductions of 4.6 Gt CO2eq by 2060 (Fig. 2). The environmental impacts of both strategies vary across material types due to differing energy intensities[28]. For example, the emission intensity of aluminum is expected to see significant declines due to the energy transition, whereas the impact on concrete is minor. As such, the effectiveness of the two strategies will reduce in the long term when energy-intensive primary metals are increasingly replaced with low-energy secondary sources[26]. This partly explains the diverging reduction potential across regions. For example, India sees a larger mitigation potential from the energy transition (61 Mt) than the China region (56 Mt) (India sees a smaller reduction when the other five strategies are implemented individually) because the latter sees a significantly higher share of secondary metals. Another reason contributing to this difference is the larger emission intensity reduction in India's material manufacturing industry from a deeper and faster energy transition.

**A High Efficiency scenario.** The High Efficiency scenario, with all material efficiency strategies (M1–M7) simultaneously applied, sees a 78 Gt CO2eq reduction (or 49%) in cumulative building-material related GHG emissions during 2020–2060 (Fig. 3). Note that the total savings from the High Efficiency scenario will not be equivalent to the aggregation of savings from each of the independent strategies because strategies can be mutually exclusive. That is, we apply these strategies (M1–M7) simultaneously and explicitly in the model framework to avoid double counting potential savings. The globally increasing trend in the Baseline scenario is reversed into a continuous decline (at an annual rate of −2.4%) during 2020–2060 (Fig. 3). Regions seeing the largest mitigation potential between this scenario and the Baseline are the China region (28%), India (16%), Western Europe (6%), Western Africa (5%), and the Middle East (5%) (in descending proportional order).

Climate targets require deep decarbonization in all sectors[59]. The building materials we consider accounted for ~7.5% of global CO2 emissions on average between 2015 and 2019. If the building material sector is to keep a share of 7.5% of the carbon budget available in this century, the HE scenario, with cumulative emissions of ~76 Gt CO2 during 2020–2060 is generally consistent with a 2 °C target (with a range of 81–144Gt at the 33–67th percentile) (see Methods). Reductions in the HE scenario are insufficient for a 1.5 °C-compatible pathway, with an emission allowance of 25–57 Gt (33–67th percentile range) during 2020–2060. Figure 3a shows the HE scenario and the trajectories stylized for the building materials sector to meet 2 °C and a 1.5 °C-compatible pathway, assuming an emission allowance of

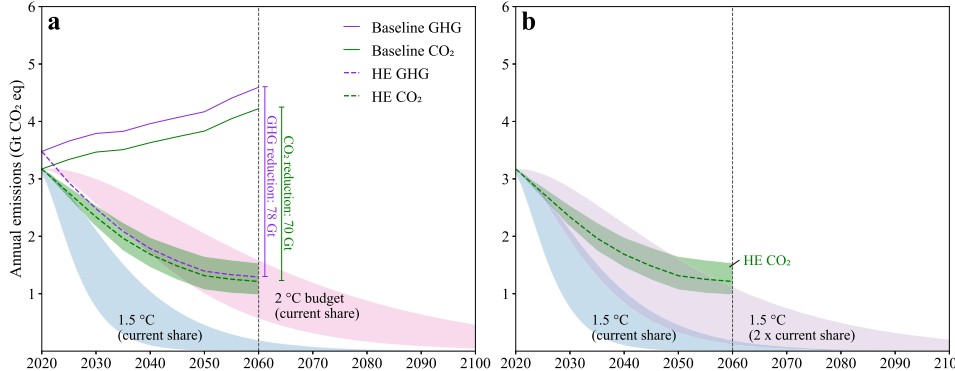

**Fig. 3 Building-material related emissions in the Baseline and High Efficiency (HE) scenarios compared with the 1.5/2 °C-compatible mitigation pathways. a** Greenhouse gas (GHG) emissions compared with the 1.5/2 °C-compatible mitigation pathways where the building material sector shares a proportional global carbon budget at 7.5%. **b** $CO_2$ emissions compared with the 1.5°C-compatible mitigation pathways where the building material sector sees a doubling share of the global carbon budget. The shaded bands in green represent the sensitivity intervals of $CO_2$ emissions in the HE scenario (as defined by 20 percentage point variations for each strategy, for more details see Supplementary Table 13). Other shaded areas represent the assessed range for the GHG emission pathways of the building material sector that are consistent with the 2 and 1.5 °C climate targets according to the IPCC, respectively, for the 33–67th percentile of TCRE (the transient climate response to cumulative carbon emissions (see Methods for details).

7.5% of the carbon budget. Figure 3b shows that for the HE scenario to be consistent with a 1.5 °C-compatible pathway the sector would require a doubling of its emission allowance. We further see that the emission reduction strategies we consider reach a saturation point around 2060 and that further strategies are needed to stay consistent with both the 1.5 and 2 °C pathways. The fact that several building materials are produced by difficult to decarbonize sectors, such as steel and cement production[60], presents a significant challenge.

There are various ways to bridge this emission reduction gap in the 1.5 °C-compatible pathway and to address the additional reductions required after 2060. First, we could assume even more ambitious versions of the strategies we investigate. However, it is questionable whether even more intensive use, further lengthening of lifetimes, and further enhancement of recycling or reuse rates are realistic. Second, we could consider other reduction strategies not included here. For example, wood cascading[61] and brick reuse[12] could reduce the use of primary materials, although compared to steel and cement these contributions would likely be small. In the material supply layer, emissions could be reduced in steel and cement production through various carbon capture, utilization, and storage technologies, such as chemical absorption[62], and calcium looping[63], among others. These technologies, and negative emission technologies which remove carbon emissions directly from the atmosphere, are still in early development and face significant technological and socio-economic barriers[64,65]. Although substantial further developments could take place up to 2060, we consider them as a complement to existing and more predictable technologies (e.g., recycling) and regulatory developments (e.g., building longevity), as broadly high-lighted in the literature[20,29]. Finally, we could assume that it is too difficult to rapidly reduce the emissions for building materials in a 1.5 °C-compatible pathway with the implication that easier-to-decarbonize sectors should realize a faster and deeper emission reduction.

**Closing material cycles.** Past decades have seen an increase in building material outflows from 1.5 Gt in 1980 to 6.5 Gt in 2020, with over 95% comprising of nonmetallic materials (especially concrete and brick) and less than 5% being metals (Supplementary Fig 3). The majority of nonmetallic outflows, except for a small fraction down-cycled as base materials, are sent as solid waste to landfills[12]. For metals, despite the already high recycling rate, inflows are much larger than outflows and primary production was still the main input

of steel (80%), copper (76%), and aluminum (69%) (over the last decade, Supplementary Fig. 3).

In the future, both outflows and inflows will be influenced by housing demand and material use strategies. On a global level, the outflow-to-inflow ratio of building materials will see a continuous increase in both Baseline and High Efficiency scenarios. The High Efficiency scenario would see a significant increase, increasing the material cycle and allowing more secondary production (Fig. 4a). However, as with other patterns, there are significant differences across regions (Fig. 4b). The potential for closing metal cycles is relatively high in high-and upper-middle-income regions that see a large in-use stock but a shrinking population such as East Asia (i.e., Japan, Korea region, and the China region), Europe, and North America, which see a steady stream of end-of-life outflow and decreasing inflow. These regions have the potential for fully closing the aluminum cycle between 2021 and 2060 under the High Efficiency scenario (Fig. 4b). By contrast, low-and lower-middle-income regions, including most African regions, South Asia, and Southeast Asia will be faced with severe scrap shortages for closing the cycles. This is not only due to the rapidly rising inflow driven mainly by population growth but also the reduced outflow from a relatively smaller in-use stock.

Some of the metal shortages in growing regions may be bridged by the surplus in shrinking regions. For example, moving surplus aluminum scrap generated in East Asia to other Asian and African regions could yield a significant reduction in the need for primary aluminum production (around 90 Mt cumulatively between 2041 and 2060), resulting in a cumulative emission reduction of ~1Gt CO2eq (in the High Efficiency scenario). It is noteworthy that China, the world's largest importer of scrap metals for many years[66], may become a major exporter in the future due to the surging outflow against shrinking inflow. In this context, China's policy restrictions on solid waste imports in recent years may be the first sign of this development[67]. Post-consumer scraps of bulk nonmetallic materials are usually processed nearby and mostly consumed by other infrastructural sectors (namely downcycling)[46]. If building demolitions are expected to be very high in certain periods then infrastructure projects should bear this in mind, reducing their requirements for primary materials and using these secondary materials. To ensure material scraps can be collected and turned into valuable resources more generally, it is important to be aware of "where and when which types of material outflows from stocks become available"[12,68,69]. Both interregional and intersectoral

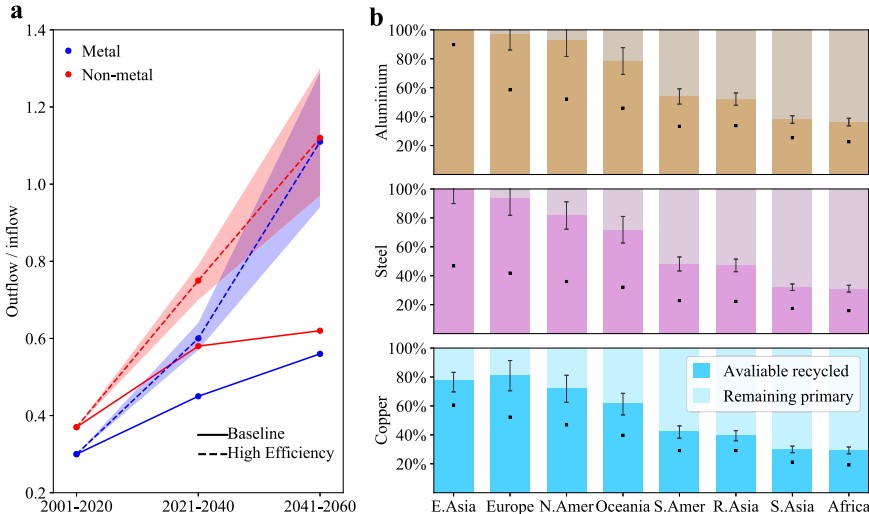

**Fig. 4 The potential for closing building material cycles. a** Change in outflow-to-inflow ratios over time (in 2001–2020, 2021–2040, and 2041–2060, respectively) under two scenarios. The shaded bands represent the sensitivity intervals of outflow-to-inflow ratios in the High Efficiency (HE) scenario (given by 20 percentage point variations for each strategy, for more details see Supplementary Table 13). **b** Share of recycled output in total input for aluminum, steel, and copper, respectively, during 2021–2060 in eight global regions (see sub-regions in the Supplementary Table 11). The whiskers represent the sensitivity intervals of the share in the HE scenario. Black dots represent the share in the Baseline scenario.

cooperation could help in urban mining and future material production capacity planning.

## Discussion

Building emissions are often complicated by trade-offs along the building lifecycle, especially between the embodied emissions (from building materials production) and operational emissions (from indoor energy use)[9,20]. Among the strategies considered in this study, more intensive use, more recovery, a faster energy transition, and production efficiency improvements are trade-off-free approaches since they don't have negative impacts on energy use during building occupation (more intensive use also reduces the operational energy use[70,71]). For lightweight design, we only consider opportunities for avoiding material overuse through improved design and technological developments, which would not compromise the building's thermal performance, so here indoor energy use will not be affected either. For material substitution by wood, previous research confirms the environmental benefits through case studies considering both the production-stage savings and potential operation-stage losses[15,72]. In terms of lifetime extension, there are concerns that older buildings tend to have lower standards so prolonging service life may increase operational energy requirements[73]. Although our analysis does not quantify this trade-off, we should highlight that such an assessment should include a longer research period (far beyond 2060) as many buildings built today will remain in use until the end of the century. On the other hand, today's buildings have generally higher energy performance compared to earlier stocks, with many recent improvements in building codes and standards (73 countries had building codes in 2018)[2,74]. This means the impact of extending the service life on energy use will be declining (even negligible in low-energy buildings). Further, much of the potential improvement in operational energy intensity lies in appliances, lighting, renewable energy, and human behavior that are not necessarily dependent on the main building structure and can be optimized at any time[75,76]. For example, in the Chinese building sector, around half of energy savings by 2050 arise from improvements in lighting, equipment and appliances, fuel switching, and renewable electricity[77]. The other half arises from space conditioning and heating, which requires both newer equipment (such as chillers) and building refurbishments (such as envelope upgrading). The environmental benefits from

building refurbishment have been reported in several case studies[21,78]. In general, the deployment of these strategies would not be hindered by trade-offs between pre-use and in-use emissions. This is not only due to the net environmental gains (over the losses) but because of the different characteristics between the embodied and operational emissions, that is, the operational emissions are generally easier to decarbonize and can often be mitigated during a building's service life.

A prominent barrier to the widespread implementation of these strategies is the fragmentation of inter-departmental policy design over time. For example, evolutionary urban planning and land policies—driven by function and/or esthetic preferences—can force a rearrangement or rezoning of the urban environment, including buildings, streets, or other infrastructure. This would increase the demolition frequency and the risk of shorter building lifetimes (in spite of their good physical condition)[51]. The lack of policy consultation between stakeholders due to political and financial interests can result in uncoordinated land urbanization and social-economic development[49,79]. This can lead to land urbanizing at a faster rate than the population, resulting in 'ghost cities' and a higher vacancy rate, especially in shrinking or population-outflow regions[79,80]. The policy options for dealing with high vacancy rates and underutilized building capacity also rely on cross-sectoral policy packages including upstream land resources management[80] and downstream taxation on vacant and rent dwellings[81]. Another example is the split incentives faced by tenants and owners in building operations. That is, those shouldering the costs of lower building efficiencies (e.g., tenants pay more for energy costs) are often those not in the position to do anything about them, which could contribute to the construction of low-quality buildings and thus frequent retrofits/demolitions. As such, policymakers are turning more towards multi-criteria decision and stakeholder-related analyses[82].

The second barrier facing these strategies is the investment required for infrastructure and technology development[19]. For example, secondary metal production can be economically and technologically challenging for large-scale alloys separation by type[38,83]. This is especially important when we consider that the proportion of emissions from high- and upper-middle-income regions may reduce as low- and lower-middle-income regions increase. This further increases the global tension between the growth in housing demand and the investment required to mitigate the

environmental impacts. As such, these strategies require coordination across regions on resource extraction, technology, and finance.

Notwithstanding these barriers, recent years have seen increasing efforts in promoting material efficiency. In terms of waste management policies, there have been several important developments within circular economy packages, such as the 3R principle (reduce, reuse, and recycle) in China[84] and the Circular Economy Action Plan (CEAP) adopted by the European Commission[85]. Strategies like light-weighting require more advanced technologies that are emerging in highly developed regions, highlighting the importance of technology marketization and international collaborations to share best practices. Similarly, higher occupation levels will likely be seen first in highly urbanized regions due to increasing vacancies from shrinking populations. The rise of a sharing economy also creates new opportunities for lower occupancy. For example, as attempted in French urban renewal projects, parking lots are shared to avoid new infrastructure construction and emissions[2].

Overall, we show that the growing housing demand drives large material-related GHG emissions which are beginning to shift from high-and upper-middle-income to low- and lower-middle-income regions. Nearly half of these emissions can be avoided through scaling up material efficiency strategies on a global level, although efficacy varies significantly with region and strategy. However, with all observed material efficiency strategies simultaneously applied, the expected emissions from building materials are still higher than what would be compatible with the 1.5 °C climate target (if the remaining global carbon budget is allocated proportionally across sectors). To meet the 1.5 °C targets, building materials would require double the current share of their carbon allowance, suggesting the need for faster emission mitigation in easier-to-decarbonize sectors. In the absence of fundamental changes in manufacturing processes, negative emissions technologies seem necessary in the second half of the century to offset process-related emissions that are challenging to avoid. This study may help policymakers to better understand the mitigation opportunities and challenges at regional and global levels and therefore how upfront investment in facilities, guidelines, and collaborations is needed.

## Methods
**Overview**. We develop an integrated global building-material-emission model that consists of a dynamic building material model and a prospective LCA model. This integrated model allows us to calculate the environmental impacts of materials used to shelter the global population and explore the impact of different material use and supply strategies on emissions. We apply this model to investigate two scenarios determined by seven key strategies in 26 global regions toward 2060 (see a conceptual framework in Supplementary Fig. 1). We include 4 residential building types (detached houses, semi-detached houses, apartments, and high-rise buildings) in urban and rural areas, respectively, and 4 commercial building types (offices, retails and warehouses, hotels and restaurants, and other commercial buildings). We include seven important construction materials: steel, concrete, brick, aluminum, copper, glass, and wood, by extending a comprehensive building material database[27,86]. IMAGE includes 26 regions (Supplementary Information), which we use as the resolution to illustrate heterogeneity in results across the globe.

**Calculation of annual material inflow and outflow**. We extend a dynamic building material assessment model (BUMA) to calculate building construction materials on a regional and yearly basis. BUMA is a cohort-based and stock-driven dynamic model, developed by Deetman et al.[27] on the basis of an open dynamic material system of Pauliuk and Heeren[87] and a floorspace model from Daioglou et al.[31]. In brief, BUMA allows for the translation from building materials stock, which is determined by socioeconomic parameters and materials use intensity of buildings, to materials inflow and outflow under a certain lifetime distribution. To do this, we derive primary socioeconomic determinants from the IMAGE platform and materials intensity from the literature. The materials intensity across global regions is collected from literature[27,86] and further developed by adding clay brick due to the extensive use of fire clay brick in buildings construction. For building lifespan, we apply Weibull distributions with related shape and scale parameters drawn from the literature[27]. Full details are provided in the Supplementary Information.

**Calculation of GHG per kg of material production**. We use a prospective LCA model to calculate GHG emissions of the production of each material type. Following the LCA procedures standardized by the International Organization for Standardization[88], we first select "cradle-to-gate" as the scope of materials production. The ecoinvent 3.6 database[32] is chosen as the lifecycle inventory (LCI) database due to its global coverage and high-resolution product categories. The regional differences in materials production are distinguished where possible. Details are shown in the Supplementary Information. We consider climate change as the key impact category, and Global Warming Potentials (with a 100-year time horizon)[89] are used. Finally, we use the activity browser (AB) software[90] to calculate the environmental impacts of the cradle-to-gate production of one kg of materials under different scenarios. The Activity Browser implements the superstructure approach[91] and significantly facilitates the modeling of future scenarios.

**Scenario development**. We investigate two scenarios that share the same socioeconomic background including population and GDP development but differ in the material intervention strategies applied. The primary socioeconomic assumptions are based on the SSPs of IMAGE and for consistency, we select the SSP2 baseline path to represent the "middle-of-the-road" pathway which expects a medium population and GDP growth[34]. In the Baseline scenario, historical trends in the building sectors around the world largely continue. We use this scenario to serve as a baseline for understanding the reduction potentials of any additional strategies. The High Efficiency scenario represents the deep emission mitigation pathway where seven strategies are implemented simultaneously. More details of the assumptions under each scenario and relevant uncertainty analysis can be found in the Supplementary Information.

**Estimation of the mitigation rate consistent with the 1.5 and 2 °C budget**. To investigate the global importance of these interventions on climate targets we also compare the Baseline and HE scenarios with stylized mitigation pathways compatible with 1.5 and 2 °C targets. Some sectors, such as electricity, are easier to decarbonize than the building material sector[60]. We, therefore, assess the efficacy of mitigation scenarios by comparing building material-related emissions against the same proportional share of the global carbon budget as today, and a situation in which the building material share doubles. We follow four steps to generate sectoral mitigation pathways consistent with the 1.5 and 2 °C carbon budgets. First, we derive the global carbon budgets from the IPCC' 1.5 °C special report[59] (see Table 2.2 in the report[59]), which indicates the remaining carbon budgets from 1/1/2018 to the time reaching net-zero carbon (or 2100) to meet the 1.5 °C Paris Agreement goal and for the former 2 °C Cancun goal. Carbon budgets here are estimated for the 33rd, 50th, and 67th percentile of TCRE (transient climate response to cumulative emissions of carbon)[92]. Second, we subtract the carbon budgets by the $CO_2$ emission in 2018 and 2019[93] to obtain the updated carbon budgets from 2020 onwards. Third, we assume the building material sector is to share the carbon budget by varying proportions. Specifically, we explore two scenarios where the building material sector shares a proportional budget of 7.5% (its average proportion of the total anthropogenic $CO_2$ emissions during 2015–2019[94]) or a doubled budget at 15.0%. We have considered $CO_2$ emission alone (representing ~92% of total GHG emissions in the sector) for this analysis since other GHGs have very different warming dynamics and comprise only a small proportion of total GHG emissions in the building material sector. Note that in practice, multiple factors (e.g., economic costs[8]) may affect sectoral effort-sharing (and therefore carbon budget allocation) in achieving a specific climate target in a period of time. Finally, we calculate mitigation rates under different carbon budgets using the method from the ref.[95] (see Eq. 4 in ref.[95]).

**Limitations and uncertainties**. While the construction-material database we use represents the best available on a global level, it could be improved to give higher geographical resolution (e.g., with national-specific and even GIS-based datasets), a higher resolution in building types, and broader coverage of material types. The materials not considered here (e.g., carpet, paint, and ceramic tiles[96]) represent further emissions on top of those examined here and potentially present different strategies for mitigation. Further, the process-based ecoinvent LCI database may underestimate some emission coefficients via truncation errors (the exclusion of small processes that are hard to quantify or those outside the defined system boundary). The future development of LCI databases for hybrid environmental flow coefficients (integrating bottom-up process data and top-down macroeconomic input-output data) may improve the completeness of assessments[91]. Another improvement of the LCI database could include accounting for the carbon sequestration effect of wood-based products using dynamic sub-models to capture the temporal effect of slow, gradual uptake of carbon in forests, along with other important factors such as the origin and rotation periods of harvesting[97]. A similar improvement could also include a dynamic sub-model to incorporate $CO_2$ reabsorption for concrete once construction is complete[25]. Finally, it is worth noting that our results are not predictions of the future but represent scenarios or pathways by which efficiency strategies can be implemented to mitigate building-material-related emissions. A sensitivity analysis (see Figs. 2–4 and the Supplementary Information for more details) is performed for understanding key interventions in the High Efficiency scenario, which further confirms both significant mitigation potentials and challenges for achieving ambitious climate goals.

## Data availability

The data that support the dynamic material and emission modeling are available from the corresponding literature references and the Supplementary Information. We have also deposited them in the Zenodo repository[98] in a form that can be easily used with our model code: https://doi.org/10.5281/zenodo.5171943. The energy system transition scenarios are not publicly available as part of the data is under license, but are available from the corresponding author upon reasonable request. Source data are provided with this paper.

## Code availability

The python code used to generate the results on material inflow, material outflow, and greenhouse gas emissions is available on Zenodo[98]: https://doi.org/10.5281/zenodo.5171943.

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

## Acknowledgements

X.Z. would like to thank the support from the China Scholarship Council (No. 201806050096).

## Author contributions

X.Z. and P.B. designed the research. X.Z. and S.D. developed the dynamic material model. X.Z., B.S., and C.H. conducted the prospective LCA modeling. X.Z. performed the analysis. X.Z. and P.B. interpreted the results. X.Z. drew the figures. X.Z. and P.B. prepared the paper. All authors contributed to discussing the results and writing the paper.

## Competing interests

The authors declare no competing interests.
