## [Peer Review File · Nature Communications]

Global greenhouse gas emissions from residential and commercial building materials and mitigation strategies to 2060
TitleREVIEWER COMMENTS

Reviewer #1 (Remarks to the Author):

This contribution is focused on a crucial GHG mitigation area, GHG emissions embodied in construction materials, and seeks to close a crucial gap in the international analysis, to develop consistent global scenarios for that area. I consider the contribution overall both informative and in many aspects well done. There are the following concerns, which I advise to be taken care of before publication can be considered:

(1) One of the main conclusions of the paper is that – in order to remain within a 1.5 degree target carbon budget - either the strategies mentioned have to be significantly further increased in strictness or other sectors have to mitigate more (both in the abstract and in lines 374-378). This leaves out an actually already pursued third option. This third option is mentioned in the article, although only for the long-term: (Lines 246-248) “Third, substantial GHG emissions can be reduced in steel and cement production through various carbon capture, utilization, and storage (CCUS) technologies, such as chemical absorption⁵⁵, calcium looping⁵⁶, and cement carbonation^{20, 57}, among others⁵⁶. These technologies are yet to be fully commercialized and may only play a significant role in reducing emission intensities in the longer term”.

(from the context one would assume that longer-term refers to basically post-2060).

However, these technological options are already available at laboratory scale, so could be employed significantly sooner than the article seems to imply. These options thus need to be discussed more inherently and considered in the scenarios, not just to be referred to as “future options”.

(2) The strategies seem to miss out significant further options. E.g. what I am told by building specialists consistently over the last years is a further significant strategy to reduce the material amount of concrete in ceilings (identically weight proof with 40-70% less concrete, and accordingly less cement), by returning to shuttering formworks as used in the early days of concrete, when it was much more expensive than labor. The new “empty segments” won could be used by activated building components, such as for storing heat (new value added chains). My suggestion is to expand for at least mentioning both such technological strategies and their integrated embedding into comprehensive overall strategies.

This is also a question of the approach. When we know that carbon neutrality is our target, such more integrative and comprehensive solutions will be sought, beyond the (somewhat narrower) strategies already mentioned in Table 1.

(3) The argument that non-CO2 emissions are hardly relevant for GHG emissions (of the building material sector) and thus a look at the carbon budget only (CO2 only) suffices, should be discussed and explained in more detail. How high is the current share/level of non-CO2 GHG emissions of this sector, and what trends could have it increase (increased relevance of insulation material), and whether we might really neglect it (as we do have a robust emissions budget for CO2 only (given the much shorter lifetime of most non-CO2 GHG emissions). But note, that non-CO2 emissions and their development do have a crucial implication on the temperature targets achieved (Figure SPM.1 of the IPCC SR 1.5 degrees).

(4) Related to that issue: How are building related GHG emissions quantified? Do you rely on a production-based accounting method, or do you – which should be the case here, at least as an alternative and possibly a sensitivity analysis – also take account of the indirect emissions, i.e. base your analysis on a consumption-based accounting approach, e.g. including emissions from electricity generation if electricity is used as an intermediate input in producing construction material. This refers to both, the current status (as given e.g. in the caption of Figure 3 – 8.1% of current GHG emissions), and in the BAU and high efficiency scenario, which both are demanding for a consumption based accounting approach. (I would expect building related emissions to account for a higher share of emissions under the consumption-based approach, which actually is a final demand allocation

approach).

(5) Further issues:

a. To avoid any potential misunderstandings you might want to add the explicit information that all your numbers in the abstract in line 22-23 refer to "building material related GHG emissions" only. Similarly, in the results section (lines 121-124).

b. For the "tradeoff between pre-use and in-use emissions whereby highly energy-efficient buildings may require more materials in construction" (lines 47-48)) there is more recent and comprehensive work then cited so far, in particular in the recent special issue of Buildings and Cities (<https://www.buildingsandcities.org/journal-content/special-issues/carbon-metrics.html>), with the trade-off explicitly discussed under a carbon budget setting in <https://doi.org/10.5334/bc.32>, which also supplies a transparent argument for larger shares of the carbon budget for certain sectors, in particular building material related GHG emissions, given the high share of investment demand in demand for such goods. You might want to consider this argument also for your purposes.

c. Lines 132-134 (and Figure 1, Panel D): that is also a matter of country size /while I understand why the absolute amount is of relevance to you here, reconsider (at least for the SI) a figure with a scaling of vertical axis rather as per capita?

d. Aggregation: Lines 151ff (and Figure 2): Mention how you aggregate these, as potentials clearly overlap.

e. Line 163ff: Note that the three layers exactly match and thus you might want to relate them to the general framework of Avoid - Shift - Improve (Creutzig et al, 2018; <https://doi.org/10.1038/s41558-018-0121-1>)

f. Typos: line 201 (infrastructure), line 441 ("that")

g. Line 234-235: "The fact that several building materials are produced by difficult to decarbonize sectors, such as steel and cement production, presents a serious challenge" – see my main points (1) and (2) above, which need to be discussed in that context here as well, them offering a clear (and earlier) way out.

Reviewer #2 (Remarks to the Author):

This is an important topic and authors have addressed it well in the paper. I would suggest its publication only after the following major revisions:

Additions:

There are indeed tradeoffs between pre-use (embodied) and in-use (operational) energy use. In fact, there are tradeoffs between in-use energy components such as heating vs. lighting or cooling loads. I suggest authors to look at the following recent study that pertains to not just the Introduction but also the Discussion section:

"Evaluating the impact of operating energy reduction measures on embodied energy"

I would suggest mentioning explicitly research objectives so that these can be mapped to the mentioned knowledge gaps and results.

I would suggest discussing construction automation technologies, particularly additive construction in the Discussion section; such technologies could have profound impact on the future housing construction.

One other barrier could be varying levels of motivation to save energy and non-energy resources

between tenants and owners. Owners tend to invest upfront to save later during the building service life, which may not be the case with tenants. Building renovation activities are not only dependent on preventive and corrective maintenance and repairs but also on other uncertain factors such as trends or fashion. These factors must also be discussed along side rezoning and reorganization of urban fabric in the Discussion section.

Clarifications:

Lines 190-192. Not a majority of intense steel-concrete structures may be substituted with timber building looking at the life cycle management aspects of timber vs. concrete/steel buildings. One big pitfall of assuming housing alone is the disregard of transportation infrastructure that will accompany any increase in housing demand. Such sectors may not be able to avoid the use of concrete and steel, at least given the current state of horizontal construction.

Lines 192-193. Timber substitution may not be feasible for all regions of the world. Regions such as India have regulations that prohibit over-exploitation of forest. If you consider the increasing housing demands in India and China alone and substitute 10% of these demands with wooden buildings, you could see the grave situation of the forests. Please make a note after this sentence so that readers can understand this aspect.

Lines 187-201. What about the increasing frequency and severity of climate change induced weather and geological disasters? Does your calculation cover post-disaster reconstruction and repairs?

Figure 4. How realistic is the High Efficiency scenario of over 0.9 of the outflow-to-inflow ratios for all material in general and for material such as wood in particular?

Overall methodological issues:

There are two major issues that authors need to address either by mentioning them as limitations or describing why these are not really important to consider. First, the time horizon of 40-80 years is too long and uncertain to predict not just the housing demand but also the type of material use that is changing rapidly with increasing use of virtual technologies and automation (e.g. 3D printed or additively constructed buildings). Does the model utilized by the authors provide any uncertainty assessment to understand these projections? Second, the energy use and energy mix are projected to change profoundly owing to automation, urbanization, and digitalization. Do you consider such changes in your model? If so, please explain. If not, please describe potential implications concerning you results.

Reviewer #3 (Remarks to the Author):

Global greenhouse gas emissions from residential and commercial building materials and mitigation strategies to 2060

General comments

=====

This paper estimates greenhouse gas emission associated with the construction and refurbishment of residential and commercial buildings, focusing on eight construction materials, from now until 2060. The paper is global in coverage and considers 26 regions around the world. The authors model seven different climate change mitigation strategies to estimate embodied greenhouse gas emissions reductions associated with construction materials. They use a process-based life cycle inventory (namely ecoinvent 3.6) to calculate embodied greenhouse gas emissions from 'cradle-to-gate' (stages A1-A3 in EN15978). The main findings demonstrate the critical need to abate embodied greenhouse gas emissions associated with construction materials, highlighting that under the best scenario, their allocated carbon budget would not be enough. A cross-sectorial approach is needed to tackled embodied greenhouse gas emissions of construction materials.

Overall, I agree with the authors on the need for this research, and I laud its global coverage and

investigation of scenarios. I do have a few suggestions to further improve the paper. These are listed below.

Firstly, the authors use ecoinvent 3.6, with some adapted versions to represent variations in energy supply chains, to calculate embodied greenhouse gas emissions. While this could be fine for a global study aiming at providing trends and comparing scenarios, the use of process data will systematically result in a truncation error in the life cycle inventory, as demonstrated by various authors in the last 20+ years (Crawford, 2008; Crawford, Bontinck, Stephan, Wiedmann, & Yu, 2018; Lenzen, 2000; Majeau-Bettez, Strømman, & Hertwich, 2011; Treloar, 1997). That means that the absolute embodied GHG figures obtained and discussed in lines 151-252, are actually even higher when using comprehensive life cycle inventory approaches such as hybrid analysis. In the only available database of hybrid embodied environmental flow coefficients for Australian construction materials (Crawford, Stephan, & Prideaux, 2019), the truncation error of ecoinvent data is more than 50% on average. I would invite the authors to at least comment on the fact that their embodied GHG estimation is truly a lower boundary in their text, as this would have even more stringent implications about 'decarbonisation' and a cross-sectorial approach.

Secondly, the authors use 8 different building types to model hundreds of millions of buildings around the world. Per m² of building modelled, I would tend to think that the 'archetypal resolution' used is rather low and this truly provides very broad-brush estimates. Recent bottom-up GIS-based studies ((Augiseau & Barles, 2017; Stephan & Athanassiadis, 2017)) have demonstrated that simply using material intensity per m² and low archetypal resolutions can lead to potentially significant errors in the quantification of the built stock, from which GHG emissions are derived. A comment on that from the authors, directly in the manuscript, would help the reader better gauge the reliability of the results. In a similar way, the authors focus on seven main materials, but previous research shows that materials outside this list can contribute significantly towards the life cycle embodied energy or greenhouse gas emissions of a building (e.g. carpet, paint, ceramic tiles, etc.)

Thirdly, in light of the above, it would be great to have some estimation of uncertainty in the final graphs produced. Only Figure 3 is visually explicit about the fact that we have bands, rather than lines. All other figures are not representing that uncertainty, which I find potentially misleading. This is a shame given that in the supplementary information, the authors have done a good sensitivity analysis, which could be represented using shaded areas, whiskers, etc. on the final graphs. I think that being very upfront about the uncertainty in this exercise would be to the credit of the authors.

Fourthly, it would be great to have some additional information on the method. The methods section does a good job at describing the procedural approach, but it might be stronger if the authors could include some further justification about their choices and their inherent limitations. For instance, choosing ecoinvent has its advantages like the authors mention, but it also results in an underestimation of the total GHG emissions. The latter is not mentioned.

Specific improvements

=====

1. Throughout: please don't use 'energy consumption'. Try 'energy use' instead. Energy cannot be consumed according to the first principle of thermodynamics.
2. In all figures, please change CO₂ to CO₂>>subscript 2<<
3. In all figure captions, please expand GHG and other abbreviations so that the capital can stand alone. If you use abbreviations such as GDP, please define these in the caption.
4. Line 194, the authors might want to cite, in addition to 33, the following paper: Arehart, J. H., Hart, J., Pomponi, F., & D'Amico, B. (2021). Carbon sequestration and storage in the built environment. *Sustainable Production and Consumption*, 27, 1047-1063.
doi:<https://doi.org/10.1016/j.spc.2021.02.028>
5. Lines 295-298: this need for more detailed information for urban mining has been called for by papers in bottom-up material stock modelling. I would suggest that the authors bring in the voices of

Tanikawa and Hashimoto (2009) and Stephan and Athanassiadis (2018) to further reinforce this argument.

6. L312, please avoid the short form, e.g. don't, in academic writing

7. L313, the authors might want to corroborate their findings on the potential of reducing floor area per capita as measure that improves environmental performance with those of previous studies on the matter, e.g. Wilson and Boehland (2005) and Stephan and Crawford (2016).

8. In the supplementary information, please check your reference entry 45, which seems to be incomplete.

9. I appreciate that the authors have shared the code and added comments across the steps. This being said, the code is extremely long and repetitive. This is just a comment and not something the authors need to act on. For the next time, it might be worth investing some time in using either vector calculations using pandas, for loops or generator expressions, to streamline the code, e.g. iterative over the list of materials and calculating relevant quantities for each.

References

=====

Augiseau, V., & Barles, S. (2017). Studying construction materials flows and stock: A review. *Resources, Conservation and Recycling*, 123, 153-164. doi:10.1016/j.resconrec.2016.09.002

Crawford, R. H. (2008). Validation of a hybrid life-cycle inventory analysis method. *Journal of Environmental Management*, 88(3), 496-506.

doi:https://www.doi.org/10.1016/j.jenvman.2007.03.024

Crawford, R. H., Bontinck, P.-A., Stephan, A., Wiedmann, T., & Yu, M. (2018). Hybrid life cycle inventory methods – a review. *Journal of Cleaner Production*, 172, 1273-1288.

doi:https://doi.org/10.1016/j.jclepro.2017.10.176

Crawford, R. H., Stephan, A., & Prideaux, F. (2019). Environmental Performance in Construction (EPiC) database. Melbourne: The University of Melbourne.

Lenzen, M. (2000). Errors in Conventional and Input-Output-based Life-Cycle Inventories. *Journal of Industrial Ecology*, 4(4), 127-148. doi:https://www.doi.org/10.1162/10881980052541981

Majeau-Bettez, G., Strømman, A. H., & Hertwich, E. G. (2011). Evaluation of process- and input-output-based life cycle inventory data with regard to truncation and aggregation issues. *Environmental Science & Technology*, 45(23), 10170-10177. doi:https://www.doi.org/10.1021/es201308x

Stephan, A., & Athanassiadis, A. (2017). Quantifying and mapping embodied environmental requirements of urban building stocks. *Building and Environment*, 114, 187-202.

doi:http://dx.doi.org/10.1016/j.buildenv.2016.11.043

Stephan, A., & Athanassiadis, A. (2018). Towards a more circular construction sector: Estimating and spatialising current and future non-structural material replacement flows to maintain urban building stocks. *Resources, Conservation and Recycling*, 129, 248-262.

doi:https://doi.org/10.1016/j.resconrec.2017.09.022

Stephan, A., & Crawford, R. H. (2016). The relationship between house size and life cycle energy demand: Implications for energy efficiency regulations for buildings. *Energy*, 116, Part 1, 1158-1171.

doi:http://dx.doi.org/10.1016/j.energy.2016.10.038

Tanikawa, H., & Hashimoto, S. (2009). Urban stock over time: spatial material stock analysis using 4d-GIS. *Building Research & Information*, 37(5-6), 483-502.

doi:https://www.doi.org/10.1080/09613210903169394

Treloar, G. J. (1997). Extracting embodied energy paths from input-output tables: towards an input-output-based hybrid energy analysis method. *Economic Systems Research*, 9(4), 375-391.

doi:https://doi.org/10.1080/09535319700000032

Wilson, A., & Boehland, J. (2005). Small is Beautiful U.S. House Size, Resource Use, and the Environment. *Journal of Industrial Ecology*, 9(1-2), 277-287.

doi:https://www.doi.org/10.1162/1088198054084680

Overall, I think that the authors have done some great work and with some additional nuancing, discussion and more transparency about uncertainty and underestimation of GHG emissions, the paper would be a significant contribution to our climate emergency.

Kind regards,
Prof. André Stephan

Detailed responses to reviewers' comments

Thank you for your email and to all reviewers for their valuable comments on our manuscript. We have carried out a comprehensive revision of the manuscript in accordance with their comments and our detailed responses to reviewers' comments are listed as follows:

R1 comments	
Reviewer's comments	Response
This contribution is focused on a crucial GHG mitigation area, GHG emissions embodied in construction materials, and seeks to close a crucial gap in the international analysis, to develop consistent global scenarios for that area. I consider the contribution overall both informative and in many aspects well done. There are the following concerns, which I advise to be taken care of before publication can be considered:	Thank you for your comments, we appreciate the constructive thoughts on how to improve the paper. Please find below the detailed responses.
(1) One of the main conclusions of the paper is that – in order to remain within a 1.5 degree target carbon budget - either the strategies mentioned have to be significantly further increased in strictness or other sectors have to mitigate more (both in the abstract and in lines 374-378). This leaves out an actually already pursued third option. This third option is mentioned in the article, although only for the long-term: (Lines 246-248) “Third, substantial GHG emissions can be reduced in steel and cement production through various carbon capture, utilization, and storage (CCUS) technologies, such as chemical absorption⁵⁵, calcium looping⁵⁶,	Thank you for this suggestion. It is worth noting that our investigations represent what-if scenarios rather than an accurate prediction of the future. These future technologies are exciting but given the level of mitigation needed and the timescales required for scaling these technologies we mainly focus on modelling material efficiency strategies that are commercially available and widely documented in recent literature (similar to other studies)^{1, 2, 3}. CCUS technologies are still at an early development stage in many areas, especially in the cement and steel sectors⁴. They still face significant technological, economic, and social barriers⁵.

and cement carbonation^{20, 57}, among others⁵⁶. These technologies are yet to be fully commercialized and may only play a significant role in reducing emission intensities in the longer term”. (from the context one would assume that longer-term refers to basically post-2060).

However, these technological options are already available at laboratory scale, so could be employed significantly sooner than the article seems to imply. These options thus need to be discussed more inherently and considered in the scenarios, not just to be referred to as “future options”.

In previous work these technologies are only considered at scale in the longer-term. For example, across IAM studies these are usually only applied at scale in the second half of the century⁶. Many also argue that the emission mitigation agenda should proceed on the premise that they will not work at scale, and that relying on the future growth of what remains a speculative technology is highly risky⁷. Therefore, alternative pathways to the 1.5 °C target are being explored to reduce⁸ or remove⁹ the dependence on negative emission efforts.

This is not to say that we will not see a faster-than-expected commercialization of these technologies, or that they are not worth significant discussion (indeed we hope that progress is rapid). Our approach is similar to many other recent modelling studies in *Science* and other venues that limit the incorporation of potential future technological breakthroughs to the discussion section^{5, 10, 11}. We have also added a further discussion on CCUS that reads:

In the material supply layer, GHG emissions could be reduced in steel and cement production through various carbon capture, utilization, and storage (CCUS) technologies, such as chemical absorption¹², calcium looping⁴, among others⁴. These technologies are yet to be fully commercialized and may only play a significant role in reducing emission intensities in the longer term. Similarly, in models reported by the IPCC, Negative Emission Technologies (NETs), those which remove

	carbon emissions directly from the atmosphere, appear at scale only in the second half of the century¹³. To avoid a reliance on untested technologies^{7, 14}, we consider them as a complement to existing and more predictable technologies (e.g. recycling) and regulatory developments (e.g. building longevity), as highlighted in the literature^{5, 15}. [see lines 261-270]
(2) The strategies seem to miss out significant further options. E.g. what I am told by building specialists consistently over the last years is a further significant strategy to reduce the material amount of concrete in ceilings (identically weight proof with 40-70% less concrete, and accordingly less cement), by returning to shuttering formworks as used in the early days of concrete, when it was much more expensive than labor. The new “empty segments” won could be used by activated building components, such as for storing heat (new value added chains). My suggestion is to expand for at least mentioning both such technological strategies and their integrated embedding into comprehensive overall strategies. This is also a question of the approach. When we know that carbon neutrality is our target, such more integrative and comprehensive solutions will be sought, beyond the (somewhat narrower) strategies already mentioned in Table 1.	Thank you for this very useful suggestion. Scenario analyses in different studies include diverse strategies to varying degrees. We agree that there is a potential for using less concrete without weakening the function of building components¹⁶ (although this may find some resistance in building regulations across the world). Following your suggestion, we have now considered concrete light-weighting in buildings as an intervention. Specifically, we assume a 10% concrete density reduction by 2050, which we feel is more feasible compared to some ambitious assumptions (e.g., 20% reduction⁵) in other literature. Similarly we have also considered the reuse of up to 15% concrete and steel^{5, 17} in the High Efficiency scenario. We have added further details in the main paper and the Supplementary Information (SI). The results have been incorporated in the final graphs, showing that the main conclusions remain robust with the newly added options.
(3) The argument that non-CO2 emissions are hardly relevant for GHG emissions (of the building material sector) and thus a look at the carbon	Thank you for pushing us to analyze the relative importance of non-CO₂ GHG emissions. We have calculated the contribution of each GHG within the LCA

budget only (CO₂ only) suffices, should be discussed and explained in more detail. How high is the current share/level of non-CO₂ GHG emissions of this sector, and what trends could have it increase (increased relevance of insulation material), and whether we might really neglect it (as we do have a robust emissions budget for CO₂ only (given the much shorter lifetime of most non-CO₂ GHG emissions). But note, that non-CO₂ emissions and their development do have a crucial implication on the temperature targets achieved (Figure SPM.1 of the IPCC SR 1.5 degrees).

analysis. Results show that CO₂ emissions represent ~91.2% of total building-material related GHG emissions (using GWP 100). This share is likely to climb slightly to ~93% in 2060, as a result of an increasing share of secondary metals (the small component of non-CO₂ GHGs, mostly CH₄, is mainly driven by primary metal production). For transparency in the related carbon budget analysis, we have considered CO₂ emissions alone. This is because other GHGs have very different warming dynamics and comprise only a small proportion of total GHGs. We have added the further explanation in the Methods, reading:

We have considered CO₂ emission alone (representing ~92% of total GHG emissions in the sector) for this analysis since other GHGs have very different warming dynamics and comprise only a small proportion of total GHGs in the building material sector (see the SI for further details).

[see lines 476-479]

We have then updated Fig. 3, as shown below:

	Fig. 3. Building-material related Greenhouse gas emissions (GHGs) by 2060 in the Baseline and High Efficiency (HE) scenarios compared with the 1.5°C/2°C-compatible mitigation pathways where (A) the building material sector shares a proportional carbon budget at 7.6% or (B) sees next to a proportional also a doubling share of the global carbon budget. The shaded bands in green represent the sensitivity intervals of CO₂ emissions in the HE scenario (as defined by 20 percentage point variations for each strategy, for more details see the SI). Other shaded areas represent the assessed range for the GHG emission pathways of the building material sector that are consistent with the 2 °C and 1.5 °C climate targets according to the IPCC, respectively, for the 33-67th percentile of TCRE (the transient climate response to cumulative emissions of carbon) (see Methods for details).
(4) Related to that issue: How are building related GHG emissions quantified? Do you rely on a production-based accounting method, or do you – which should be the case here, at least as an alternative and possibly a sensitivity analysis – also take account of the indirect emissions, i.e. base your analysis on a consumption-based accounting approach, e.g. including emissions from electricity generation if electricity is used as an intermediate input in producing construction material. This refers to both, the current status (as given e.g. in the caption of Figure 3 – 8.1% of current GHG emissions), and in the BAU and high efficiency scenario, which both are demanding for a consumption based accounting approach.	Thank you for requesting further clarifications. First, we define the building-material related emissions as emissions induced by material production. Therefore, on-site construction activities are out of scope for this assessment. We use ecoinvent 3.6, a process-based life cycle inventory (LCI) to quantify the GHGs driven by materials production from ‘cradle-to-gate’, i.e., from the ore extraction, the collection of scrap to the produced primary or secondary materials. The indirect emissions, e.g., from intermediate electricity generation to produce these materials, are also considered. Different types of LCI (e.g, process, input-output, or hybrid LCIs) include intermediate processes to varying extents and face different trade-offs for use in

(I would expect building related emissions to account for a higher share of emissions under the consumption-based approach, which actually is a final demand allocation approach).

assessment. Process-based LCI analyses are very common as they often have the best sectoral resolution and have high reliability for the core processes needing characterization, but they have the potential to exclude some smaller processes that are hard to quantify or outside the defined system boundary (termed a truncation error)^{18, 19}. In method, our approach is very close to Clark et al, 2020¹¹ (a recent article on 1.5-degree emissions from the food system in *Science*) who uses process-based LCIs of food products. We use process-based ecoinvent emission inventories with global coverage and regional resolution, offering reliable cradle-to-gate emission coefficients for major construction materials. Importantly, we link these inventories with IAMs to reflect the influences of technological and social-economic changes (e.g., technological developments in electricity used for intermediate material processes). To help readers understand these distinctions we have added the further discussion in the Methods:

Further, the process-based ecoinvent LCI database may underestimate some emission coefficients via truncation errors (the exclusion of small processes that are hard to quantify or those outside the defined system boundary). The future development of LCI databases for hybrid environmental flow coefficients (integrating bottom-up process data and top-down macroeconomic input-output data) may improve the completeness of assessments²⁰.

[see lines 491-494]

Another potential underestimation includes the exclusion of some construction

	materials, e.g., carpet, paint, and ceramic tiles²⁰. We have also added a discussion about this in the manuscript: While the construction-material database we use represents the best available on a global level, it could be improved to give higher geographical resolution (e.g., with national-specific and even GIS-based datasets), a higher resolution in building types, and a broader coverage of material types. The materials not considered here (e.g., carpet, paint, and ceramic tiles²⁰) represent further emissions on top of those examined here and potentially present different strategies for mitigation. [see lines 487-491] As for the emission share of the sector, the literature²¹ indicates that building materials represent ~11% of energy-related CO₂ emissions (~3.6 Gt), which equals ~8.5% of total anthropogenic CO₂ emissions. This is only slightly higher than our estimate for today (~3.3 Gt) which may be explained by slightly different production boundaries and material categories (as mentioned above).
(5) Further issues: a. To avoid any potential misunderstandings you might want to add the explicit information that all your numbers in the abstract in line 22-23 refer to “building material related GHG emissions” only. Similarly, in the results section (lines 121-124). b. For the “tradeoff between pre-use and in-use emissions whereby highly energy-efficient buildings may require more materials in	Thank you for highlighting these points. We have made revisions accordingly: a. We have added the explicit meaning of the emissions by indicating “building-material related GHG emissions” in both Abstract and Results. [see lines 24 and 125] b. Thank you for highlighting this. We have now cited Steininger et al. (2020)²² in the Introduction (lines 48-50) and have added further discussion of their insights on sectoral carbon budget evaluation, which reads:

construction” (lines 47-48)) there is more recent and comprehensive work then cited so far, in particular in the recent special issue of Buildings and Cities (<https://www.buildingsandcities.org/journal-content/special-issues/carbon-metrics.html>), with the trade-off explicitly discussed under a carbon budget setting in, which also supplies a transparent argument for larger shares of the carbon budget for certain sectors, in particular building material related GHG emissions, given the high share of investment demand in demand for such goods. You might want to consider this argument also for your purposes.

- c. Lines 132-134 (and Figure 1, Panel D): that is also a matter of country size /while I understand why the absolute amount is of relevance to you here, reconsider (at least for the SI) a figure with a scaling of vertical axis rather as per capita?
- d. Aggregation: Lines 151ff (and Figure 2): Mention how you aggregate these, as potentials clearly overlap.
- e. Line 163ff: Note that the three layers exactly match and thus you might want to relate them to the general framework of Avoid - Shift - Improve (Creutzig et al, 2018; <https://doi.org/10.1038/s41558-018-0121-1>)
- f. Typos: line 201 (infrastructure), line 441 (“that”)
- g. Line 234-235: “The fact that several building materials are produced by difficult to decarbonize sectors, such as steel and cement

Note that in practice, multiple factors (e.g., economic costs²²) may impact sectoral effort-sharing (and therefore carbon budget allocation) to achieve a specific climate target.

[see lines 479-481]

c. Thank you for this suggestion. To exclude the influence of the regional economy size, we have updated Fig. 1D to demonstrate the cumulative GHG emissions per GDP, as shown below.

Fig. 1. Greenhouse gas emissions (GHGs) from building materials use

production, presents a serious challenge” – see my main points (1) and (2) above, which need to be discussed in that context here as well, them offering a clear (and earlier) way out.

for global regions in the baseline scenario. (A) Development of global GHGs for 7 materials during 2020-2060. (B) Percentage evolution of GHGs for three income groups during 2000-2060. (C) Development of emissions in the top 6 emitting regions (by 2060), occupying over 60% of the total, during 2020-2060. (D) Expected cumulative GHGs over 2020-2060 relative to present GDP (2020 value from the IMAGE integrated model, at purchasing power parity) for 26 global regions. The updated wording in the main text reads:

Figure 1D shows the regional comparison of cumulative material-related GHGs relative to GDP, highlighting contrasting economic challenges for the adoption of mitigation strategies. In general, high-income regions (such as the US, Japan, and Western Europe) will see relatively lower emissions and, therefore, have higher affordability of deep decarbonization.

[see lines 135-139]

d. Thank you for pushing us to clarify. We first give the impacts of each strategy independently (see Fig. 2). For example, when investigating the potential of energy transition, we replace the Baseline energy database (SSP2-baseline) with the High-efficiency energy database (SSP2-2.6) in the LCA background system, while all other variables remain the same as in the Baseline version. We have briefly indicated this in the manuscript, reading:

Figure 2 shows the reduction potential for each strategy at their High Efficiency levels during 2021-2060 (in comparison with the Baseline values and when each

strategy is adopted independently of one another).

[see lines 162-164]

As you note, it is not sensible to aggregate all these mitigation potentials since strategies clearly overlap and their full potentials are mutually exclusive. To explore the total potential of employing all strategies, we simultaneously apply all strategies (i.e., M1 to M7) at once in the High Efficiency scenario in the modeling system. That is, the application of all strategies is represented in a non-mutually exclusive way explicitly in the model. In short, the emission savings from the High Efficiency scenario is not equivalent to the aggregation of savings from applying the strategy M1 to M7. The results are demonstrated in Fig 3. Similar approaches are commonly seen in recent studies to observe pollution reduction potential of multiple interventions (and their combinations), for example, in the food sector¹¹, cement sector¹⁰, and aluminium sector²³. We have updated the text in the paper to read:

The High Efficiency scenario, with all material efficiency strategies (M1-M7) simultaneously applied, sees a cumulative 81 Gt CO₂eq reduction (or 49%) in building-material related GHG emissions during 2020-2060 (Fig. 3). Note that the total savings from the High Efficiency scenario will not be equivalent to the aggregation of savings from each of the independent strategies because strategies can be mutually exclusive. That is, we apply these strategies (M1-M7) simultaneously and explicitly in the model framework to avoid double counting potential savings.

	[see lines 231-236] e. We have related these three layers to the avoid–shift–improve framework. The revised text reads: The three colors left to right represent the three layers in the modelling framework: building demand, material demand, and material supply (see Supplementary Figure S.1). These three approaches correspond approximately to the general ‘avoid–shift–improve’ emission mitigation framework²⁴. [see lines 164-166] f. Thank you for highlighting this. We have corrected the typos in line 201 (‘infrastructure’) and line 441 (‘that’). We have also double-checked the text throughout the manuscript to avoid typos. g. Thank you; we have now addressed this issue in your comments 1) and 2).
--	---

R2 comments

Reviewer’s comments	Response
This is an important topic and authors have addressed it well in the paper. I would suggest its publication only after the following major revisions:	Thank you for your comments and useful suggestions. Please find below the detailed responses (numbered as 2.1-2.10 for ease of inspection).
Additions: 2.1 There are indeed tradeoffs between pre-use (embodied) and in-use (operational) energy use. In fact, there are tradeoffs between in-use energy	2.1 Thank you for pointing us to Venkatraj et al. (2020)²⁵. We have added further discussion on this in both Introduction and Discussion with the revised text reading:

components such as heating vs. lighting or cooling loads. I suggest authors to look at the following recent study that pertains to not just the Introduction but also the Discussion section: “Evaluating the impact of operating energy reduction measures on embodied energy”	There may also be a tradeoff between pre-use and in-use emissions whereby highly energy-efficient buildings may require more materials in construction^{22, 25, 26}. [see lines 46-48] The emissions from buildings are often complicated by trade-offs along the lifecycle of a building, especially between the embodied emissions (from building materials production) and operational emissions (from indoor energy use)^{1, 3, 25}. [see lines 308-310]
2.2 I would suggest mentioning explicitly research objectives so that these can be mapped to the mentioned knowledge gaps and results.	2.2 Thank you for pushing us to clearly describe the research objectives. We have added the objectives in the Introduction, reading: Here we investigate the development of global GHG emissions of residential and commercial building material production. We investigate, the impacts of major material efficiency strategies, and the implications of these strategies for on meeting world climate targets. [see lines 62-64]
2.3 I would suggest discussing construction automation technologies, particularly additive construction in the Discussion section; such technologies could have profound impact on the future housing construction.	2.3 Thank you for this suggestion. From a material efficiency point of view, additive construction technologies such as 3D printing may reduce the amount of material use since components are printed on-demand and with high accuracy. This is one of the many techniques that help for light weighting. At the same time, technologies such as prefabrication and modular construction

may also yield the potential for increased material reuse. The light weighting scenario we considered (up to 19% reduction for steel and aluminium) is analogous to the impact of various different future technologies and design approaches. On top of that, to consider these technologies more comprehensively, we have now added a 10% light weighting potential for concrete and 15% reuse potential for after-use concrete and steel (see the manuscript and SI for more details). The updated wording in the SI now reads:

Current construction practices tend to overuse materials due to less efficient design or a relatively low physical strength of primary materials^{17, 27}. There is a potential for lightweighting through more advanced design strategies including novel structural design²⁸, typology optimization²⁹, additive construction (such as 3D printing)³⁰, and high strength steel and aluminium utilization³¹.

[see lines 145-148 in the SI]

There is a potential for increasing the reuse of building components through increased prefabrication and modular construction design⁵. Based on case studies, we assume that up to 15%¹⁷ of steel and concrete could be reused with these approaches. Both assumptions are somewhat conservative compared to the recent scenario analysis for several countries⁵, yet were chosen considering the varying technology and regulation development patterns across global regions. In the Baseline scenario, no reuse is adopted for all materials. In the High efficiency scenario, we assume a linear increase to 15%

	reuse for concrete and steel between 2020 to 2050 and hold constant afterwards. [see lines 209-215 in the SI]
2.4 One other barrier could be varying levels of motivation to save energy and non-energy resources between tenants and owners. Owners tend to invest upfront to save later during the building service life, which may not be the case with tenants. Building renovation activities are not only dependent on preventive and corrective maintenance and repairs but also on other uncertain factors such as trends or fashion. These factors must also be discussed along side rezoning and reorganization of urban fabric in the Discussion section.	2.4 Thank you for this interesting point on split incentives. We have added a short discussion in the manuscript, which now reads: Another example is the split-incentives faced by tenants and owners in building operation. That is, those shouldering the costs of lower building efficiencies (e.g., tenants pay more for energy costs) are often those not in the position to do anything about them, which could contribute to the construction of low-quality buildings and frequent retrofits/demolitions. [see lines 387-391] We have then incorporated building trends and aesthetic concerns in the discussion. The updated wording reads: For example, evolutionary urban planning and land policies – driven by function and/or aesthetic preferences – can force a rearrangement or rezoning of the urban environment, including buildings, streets, or other infrastructure. This would increase demolition frequency and the risk of shorter building lifetimes (in spite of their good physical condition)³². [see lines 375-377]
Clarifications: 2.5 Lines 190-192. Not a majority of intense steel-concrete structures may	Thank you for this comment. A number of studies have explored the potential for timber substitution¹. For example, Churkina et al³³ explored

be substituted with timber building looking at the life cycle management aspects of timber vs. concrete/steel buildings. One big pitfall of assuming housing alone is the disregard of transportation infrastructure that will accompany any increase in housing demand. Such sectors may not be able to avoid the use of concrete and steel, at least given the current state of horizontal construction.	a potential substitution of up to 90% of global mid-rise buildings. We take a much more conservative scenario in assuming a 10% substitution (due to land-use and forestry concerns regarding other sustainability issues such as biodiversity^{34, 35}). This is also relevant to your next comment.
2.6 Lines 192-193. Timber substitution may not be feasible for all regions of the world. Regions such as India have regulations that prohibit over-exploitation of forest. If you consider the increasing housing demands in India and China alone and substitute 10% of these demands with wooden buildings, you could see the grave situation of the forests. Please make a note after this sentence so that readers can understand this aspect.	Thank you for pushing us to clarify and we agree. We have added further text reading: Globally 10% timber substitution as a high-efficiency strategy is rather conservative (others having investigated 90% substitution scenarios³³). We choose this lower substitution level due to concerns surrounding land use change to managed forests and related biodiversity loss³⁴, especially in rapidly growing Asian and African regions. [see lines172-176 in the SI]
2.7 Lines 187-201. What about the increasing frequency and severity of climate change induced weather and geological disasters? Does your calculation cover post-disaster reconstruction and repairs?	While this is very interesting it is currently out of scope of this paper. Indeed, there are still many methodological issues with making such an analysis (see https://www.politico.com/news/2021/03/16/climate-change-murky-models-476316 for a popular overview). It is a crucial area of further research. We have added a discussion in the SI, which now reads: Note that although we mainly consider social-economic factors (i.e., population

	and floor area per person) as the driver of the global demand in building stocks. There are other factors that will impact future building stocks and could be modelled in future work (using different modelling approaches). For example, the increasing frequency and severity of climate changes may create extra building demand as a complement for those damaged by natural disasters³⁶, or abandoned in areas declared uninhabitable³⁷. [see lines 57-61 in the SI]
2.8 Figure 4. How realistic is the High Efficiency scenario of over 0.9 of the outflow-to-inflow ratios for all material in general and for material such as wood in particular?	Firstly, it is worth noting that our analyses represent ‘what-if’ scenarios rather than a prediction of the future. The ratio over 0.9 for various materials, for example, is a result of applying the different interventions so is an emergent result from the application of M1-M4. We do not suggest that this will be easy but that it is feasible due to the interplay of these different interventions based on the literature. Secondly, there are also scenarios from other literature that suggest the potential for (nearly-)fully closing material circles such as steel³⁸ and aluminium²³ by around the mid-century. At the same time, there are some signs that aspirations to achieve such circularity are increasing. For example, the Netherlands has a 100% circular economy target for 2050 (https://www.government.nl/documents/policy-notes/2016/09/14/a-circular-economy-in-the-netherlands-by-2050) and the EU also launched ambitious circular economy target for 2050 (https://eur-lex.europa.eu/legal-content/EN/TXT/?uri=COM:2020:98:FIN)

Overall methodological issues:

There are two major issues that authors need to address either by mentioning them as limitations or describing why these are not really important to consider.

2.9 First, the time horizon of 40-80 years is too long and uncertain to predict not just the housing demand but also the type of material use that is changing rapidly with increasing use of virtual technologies and automation (e.g. 3D printed or additively constructed buildings). Does the model utilized by the authors provide any uncertainty assessment to understand these projections?

Thank you for requesting further clarification about the limitations/uncertainties about the scenarios. We agree that scenarios over such long time can have a lot of uncertainties. However, our scenarios actually cover a shorter time horizon than many other scenarios in the literature. For example, the IMAGE model, on which we base our Baseline scenario, and other integrated assessment models (IAMs) usually cover a much longer time period (usually to 2100)^{39,40}. For example, Leclère et al., 2020⁴¹ in *Nature* explore future global biodiversity trends resulting from land-use change to 2100 using IMAGE and other IAMs. Similarly, Luderer et al., 2018⁴² in *Nature Climate Change* analyse the residual fossil CO₂ emissions to 2100 using IMAGE and other IAMs. There are many such examples. In the High Efficiency scenario, we mainly focus on modelling technologies that are commercially available today and widely documented in the literature. Since housing does have a long capital stock turnover, we can expect that any interesting and new technologies not commercially available today will take time to scale and then diffuse through the sector. However, we have broadly included technologies such as a 3-D printing and additive construction in the light-weighting and reuse approach (as you suggested in comment 2.3 above).

At the same time, we acknowledge the limitations and uncertainties in our analysis. Another reviewer also suggested a sensitivity analysis and

	further discussion about the limitations. Considering both, we have first conducted a sensitivity analysis about the change in key material efficiency strategies, results of which have been shown in the final graphs (see Fig 2-4). We have also added a ‘Limitations and uncertainties’ discussion in the manuscript. Specifically, we have discussed about the limitations in both modelling and data, as well as a comparison between several variables in our study and the literature.
2.10 Second, the energy use and energy mix are projected to change profoundly owing to automation, urbanization, and digitalization. Do you consider such changes in your model? If so, please explain. If not, please describe potential implications concerning your results.	By basing electricity scenarios on the “The Image Energy Regional Model” (TIMER)^{39, 43}, our analysis considers many developments in the electricity system (from both electricity demand and supply sides). TIMER represents integrated scenarios of market share of various competing energy technologies (e.g., solar, wind, hydro, nuclear, biomass, and fossil fuels) for different SSPs. A full list of these technologies and scenarios can be found in the Supporting Information of Mendoza Beltran et al., 2020⁴³. Following your suggestion, we have added a brief description of these aspects in the SI. The updated text reads: These electricity scenarios have incorporated consistent techno socio-economic development in both energy demand (e.g., population size, income-level, and lifestyle) and energy supply (e.g., costs of competing electricity

generating technologies) determinants⁴³. Specifically, we apply the electricity scenarios that are compatible with SSP2-baseline and SSP2-2.6 pathways to our Baseline and High Efficiency scenarios, respectively.

[see lines 226-230 in the SI]

R3 comments

Reviewer's comments	Response
General comments This paper estimates greenhouse gas emission associated with the construction and refurbishment of residential and commercial buildings, focusing on eight construction materials, from now until 2060. The paper is global in coverage and considers 26 regions around the world. The authors model seven different climate change mitigation strategies to estimate embodied greenhouse gas emissions reductions associated with construction materials. They use a process-based life cycle inventory (namely ecoinvent 3.6) to calculate embodied greenhouse gas emissions from 'cradle-to-gate' (stages A1-A3 in EN15978). The main findings demonstrate the critical need to abate embodied greenhouse gas emissions associated with construction materials, highlighting that under the best scenario, their allocated carbon budget would not be enough. A cross-sectorial approach is needed to tackled embodied greenhouse gas emissions of construction materials. > Overall, I agree with the authors on the need for this research, and I laud its global coverage and investigation of scenarios. I do have a few suggestions to further improve the paper. These are listed below.	Thank you for your comments. Please find below the detailed responses (numbered as 3.1-3.5 for ease of inspection).
3.1 Firstly, the authors use ecoinvent 3.6, with some adapted versions to	3.1 Thank you for highlighting this. Indeed, the hybrid approach is likely to

represent variations in energy supply chains, to calculate embodied greenhouse gas emissions. While this could be fine for a global study aiming at providing trends and comparing scenarios, the use of process data will systematically result in a truncation error in the life cycle inventory, as demonstrated by various authors in the last 20+ years (Crawford, 2008; Crawford, Bontinck, Stephan, Wiedmann, & Yu, 2018; Lenzen, 2000; Majeau-Bettez, Strømman, & Hertwich, 2011; Treloar, 1997). That means that the absolute embodied GHG figures obtained and discussed in lines 151-252, are actually even higher when using comprehensive life cycle inventory approaches such as hybrid analysis. In the only available database of hybrid embodied environmental flow coefficients for Australian construction materials (Crawford, Stephan, & Prideaux, 2019), the truncation error of ecoinvent data is more than 50% on average. I would invite the authors to at least comment on the fact that their embodied GHG estimation is truly a lower boundary in their text, as this would have even more stringent implications about ‘decarbonisation’ and a cross-sectorial approach.

improve the accuracy of the LCA result mainly by reducing truncation errors. However, hybrid approaches have seen slow adoption in analyses due to several key challenges (Crawford et al., 2018)¹⁸. Hybrid analyses experience reduced precision in their results due to the wider distribution of input-output data (Perkins and Suh, 2019)¹⁹. Additionally, there is a lack of consistency among hybrid methods that may lead to significantly different LCA results (Crawford et al., 2018)¹⁸. Overall, we agree that hybrid approaches represent a promising opportunity for improvement and that the Australian database (Crawford et al., 2019)²⁰ is a good starting point. As we noted in a response to reviewer 1 we use a very similar methodology to the one used by Clark et al, 2020¹¹ for the food system (recently published in *Science*). Following your suggestion, we have added a discussion on potential underestimations from using process-based life cycle inventories (LCI), which reads:

Further, the process-based ecoinvent LCI database may underestimate some emission coefficients via truncation errors (the exclusion of small processes that are hard to quantify or those outside the defined system boundary). The future development of LCI databases for hybrid environmental flow coefficients (integrating bottom-up process data and top-down macroeconomic input-output data) may improve the completeness of assessments²⁰.

[see lines 497-502]

3.2 Secondly, the authors use 8 different building types to model

3.2 Thank you for this suggestion. We have now included comments on

hundreds of millions of buildings around the world. Per m² of building modelled, I would tend to think that the ‘archetypal resolution’ used is rather low and this truly provides very broad-brush estimates. Recent bottom-up GIS-based studies ((Augiseau & Barles, 2017; Stephan & Athanassiadis, 2017)) have demonstrated that simply using material intensity per m² and low archetypal resolutions can lead to potentially significant errors in the quantification of the built stock, from which GHG emissions are derived. A comment on that from the authors, directly in the manuscript, would help the reader better gauge the reliability of the results. In a similar way, the authors focus on seven main materials, but previous research shows that materials outside this list can contribute significantly towards the life cycle embodied energy or greenhouse gas emissions of a building (e.g. carpet, paint, ceramic tiles, etc.)

this in the manuscript. This updated wording reads:

While the construction-material database we use represents the best available on a global level, it could be improved to give higher geographical resolution (e.g., with national-specific and even GIS-based datasets), a higher resolution in building types, and a broader coverage of material types. The materials not considered here (e.g., carpet, paint, and ceramic tiles²⁰) may represent considerable GHGs and different strategies for mitigation.

[see lines 493-497]

3.3 Thirdly, in light of the above, it would be great to have some estimation of uncertainty in the final graphs produced. Only Figure 3 is visually explicit about the fact that we have bands, rather than lines. All other figures are not representing that uncertainty, which I find potentially misleading. This is a shame given that in the supplementary information, the authors have done a good sensitivity analysis, which could be represented using shaded areas, whiskers, etc. on the final graphs. I think that being very upfront about the uncertainty in this

Thank you for pushing us to highlight the sensitivity intervals in the final graphs. Following your suggestion, we have added shaded bands or whiskers on Fig 2-4, as shown below:

exercise would be to the credit of the authors.

Fig. 2. Greenhouse gas emission (GHG) mitigation potential during 2021-2060 by different material efficiency strategies. The three colors left to right represent the three layers in the modelling framework: building demand, material demand, and material supply (see Supplementary *Figure S.1*). These three approaches correspond approximately to the general ‘avoid–shift–improve’ emission mitigation framework²⁴. The whiskers represent the sensitivity intervals of GHGs in the HE scenario (given by 20 percentage point variations for each strategy; see SI for further details). Note that the scales for Global, the China region, and India differ from other regions, and the scale for more intensive use differs from other strategies.

Fig. 3. Building-material related Greenhouse gas emissions (GHGs) by 2060 in the Baseline and High Efficiency (HE) scenarios compared with the 1.5°C/2°C-compatible mitigation pathways where (A) the building material sector shares a proportional carbon budget at 7.6% or (B) sees next to a proportional also a doubling share of the global carbon budget. The shaded bands in green represent the sensitivity intervals of CO₂ emissions in the HE scenario (as defined by 20 percentage point variations for each strategy, for more details see the SI). Other shaded areas represent the assessed range for the GHG emission pathways of the building material sector that are consistent with the 2 °C and 1.5 °C climate targets according to the IPCC, respectively, for the 33-67th percentile of TCRE (the transient climate response to cumulative emissions of carbon) (see Methods for details).

Fig. 4. The potential for closing building material cycles. (A) Change in outflow-to-inflow ratios over time (in 2000-2020, 2021-2040, and 2041-2060, respectively) under two scenarios. The shaded bands represent the sensitivity intervals of outflow-to-inflow ratios in the HE scenario (as described above and in the SI). (B) Share of recycled output in total input for aluminium, steel, and copper, respectively, during 2021-2060 in eight global regions (see sub-regions in the Supplementary Table S.11). The whiskers represent the sensitivity intervals of the share in the HE scenario. Black dots represent the share in the Baseline scenario.

3.4 Fourthly, it would be great to have some additional information on

Thank you for this suggestion. First, we have now added a ‘Limitations

the method. The methods section does a good job at describing the procedural approach, but it might be stronger if the authors could include some further justification about their choices and their inherent limitations. For instance, choosing ecoinvent has its advantages like the authors mention, but it also results in an underestimation of the total GHG emissions. The latter is not mentioned.

and uncertainties' discussion in the manuscript. The choice of using ecoinvent, for instance, is discussed there as you suggested. See details in the main text and the SI.

Second, we have also added our reflections on primary assumptions. For example, we assume the lifetime of buildings follows a Weibull distribution. While this is common in the literature, other options have also been suggested. The updated text in the SI now reads:

Note that the functional form for the lifetime distributions can have a considerable impact on the calculation of outflow and inflow⁴⁴. Other functional forms, e.g., normal⁴⁵, Log-normal⁴⁶, and Gamma functions⁴⁷ have also used in material related dynamic analysis. However, the Weibull distribution is the most common in the literature, and is therefore the distribution for which most reliable data are available⁴⁸.

[see lines 43-47 in the SI]

Another example is the degradation over time of reused components that may lose some of their function during the service life of the building. We assume that up to 2060 the materials will still be functional as reused components. However, for a longer time period (e.g., beyond 2100), split lifetimes may need to be considered between the buildings and the reused components. The added wording reads:

Note that an inherent assumption is that reused components will meet the

	function needed during the service life of the building. While this may be quite reasonable in our scenarios up to 2060, it may not be so for later in the century. Split lifespans of the buildings and reused components may be considered when exploring scenarios of longer time horizon, e.g., towards and beyond 2100. [see lines 215-219 in the SI] See more reflection details in the main text and the SI.
3.5 Specific improvements 1. Throughout: please don't use 'energy consumption'. Try 'energy use' instead. Energy cannot be consumed according to the first principle of thermodynamics. 2. In all figures, please change CO₂ to CO₂ 3. In all figure captions, please expand GHG and other abbreviations so that the capital can stand alone. If you use abbreviations such as GDP, please define these in the caption. 4. Line 194, the authors might want to cite, in addition to 33, the following paper: Arehart, J. H., Hart, J., Pomponi, F., & D'Amico, B. (2021). Carbon sequestration and storage in the built environment. Sustainable Production and Consumption, 27, 1047-1063. doi:https://doi.org/10.1016/j.spc.2021.02.028 5. Lines 295-298: this need for more detailed information for urban mining has been called for by papers in bottom-up material stock	We very much appreciate the time you have taken to catch these improvements. We have addressed all the points below:  1. We have replaced 'energy consumption' by 'energy use' throughout the manuscript. 2. We have changed CO₂ to CO₂ in all figures. 3. We have expanded 'GHG' to 'greenhouse gas emission' the first time we use it in the caption of a figure such that other 'GHG' in the same figure caption can be understood. We have defined GDP as gross domestic product in the caption of Figure 1. 4. We have cited the paper of Arehart et al., 2021⁴⁹. The updated sentence reads: A 10% substitution with timber buildings by 2060 levels would result in GHG emission reduction of 3.9 Gt CO₂eq (due to the lower emission intensity of timber production) and provide long-term carbon storage^{33, 49}.

modelling. I would suggest that the authors bring in the voices of Tanikawa and Hashimoto (2009) and Stephan and Athanassiadis (2018) to further reinforce this argument.

6. L312, please avoid the short form, e.g. don't, in academic writing

7. L313, the authors might want to corroborate their findings on the potential of reducing floor area per capita as measure that improves environmental performance with those of previous studies on the matter, e.g. Wilson and Boehland (2005) and Stephan and Crawford (2016).

8. In the supplementary information, please check your reference entry 45, which seems to be incomplete.

9. I appreciate that the authors have shared the code and added comments across the steps. This being said, the code is extremely long and repetitive. This is just a comment and not something the authors need to act on. For the next time, it might be worth investing some time in using either vector calculations using pandas, for loops or generator expressions, to streamline the code, e.g. iterative over the list of materials and calculating relevant quantities for each.

References

>Augiseau, V., & Barles, S. (2017). Studying construction materials flows and stock: A review. *Resources, Conservation and Recycling*, 123, 153-164. doi:10.1016/j.resconrec.2016.09.002

> Crawford, R. H. (2008). Validation of a hybrid life-cycle inventory

[see lines 207-209]

5. We have cited both Tanikawa and Hashimoto (2009) and Stephan and Athanassiadis (2018). The updated sentences read:

To ensure material scraps can be collected and turned into valuable resources more generally, it is important to be aware of “where and when which types of material outflows” from stocks become available^{50, 51, 52}. Both interregional and intersectoral cooperation could help in urban mining and future material production capacity planning.

[see lines 327-331]

6. We have replaced ‘don’t’ by ‘do not’. We have also double-checked the writing throughout the manuscript to avoid using any informal short abbreviations.

7. Thank you bringing this reference to our attention. We have now cited the work of Wilson and Boehland (2005)⁵³ and Stephan and Crawford (2016)⁵⁴ for cross-checking. The updated sentence reads:

Among the strategies considered in this study, more intensive use, more recycling, a faster energy transition, and production efficiency improvements are trade-off-free approaches since they don’t have negative impacts on energy use during building occupation (more intensive use also reduces the operational energy use^{53, 54}).

[see lines 346-349]

analysis method. *Journal of Environmental Management*, 88(3), 496-506. doi:<https://www.doi.org/10.1016/j.jenvman.2007.03.024>

> Crawford, R. H., Bontinck, P.-A., Stephan, A., Wiedmann, T., & Yu, M. (2018). Hybrid life cycle inventory methods – a review. *Journal of Cleaner Production*, 172, 1273-1288. doi:<https://doi.org/10.1016/j.jclepro.2017.10.176>

> Crawford, R. H., Stephan, A., & Prideaux, F. (2019). *Environmental Performance in Construction (EPiC) database*. Melbourne: The University of Melbourne.

> Lenzen, M. (2000). Errors in Conventional and Input-Output-based Life-Cycle Inventories. *Journal of Industrial Ecology*, 4(4), 127-148. doi:<https://www.doi.org/10.1162/10881980052541981>

> Majeau-Bettez, G., Strømman, A. H., & Hertwich, E. G. (2011). Evaluation of process- and input-output-based life cycle inventory data with regard to truncation and aggregation issues. *Environmental Science & Technology*, 45(23), 10170-10177. doi:<https://www.doi.org/10.1021/es201308x>

> Stephan, A., & Athanassiadis, A. (2017). Quantifying and mapping embodied environmental requirements of urban building stocks. *Building and Environment*, 114, 187-202. doi:<http://dx.doi.org/10.1016/j.buildenv.2016.11.043>

> Stephan, A., & Athanassiadis, A. (2018). Towards a more circular construction sector: Estimating and spatialising current and future non-

8. We have updated the reference entry of ref 45 in the SI. It now reads:

World Aluminium. Statistics. <http://www.world-aluminium.org/statistics/>. Accessed December 2016.

We have also double-checked the reference list in both the main paper and SI to avoid any invalid entries.

9. Thank you for the suggestions on coding techniques. We will spend more time in improving the code for efficiency and ease of reading.

structural material replacement flows to maintain urban building stocks. *Resources, Conservation and Recycling*, 129, 248-262.
doi:<https://doi.org/10.1016/j.resconrec.2017.09.022>

> Stephan, A., & Crawford, R. H. (2016). The relationship between house size and life cycle energy demand: Implications for energy efficiency regulations for buildings. *Energy*, 116, Part 1, 1158-1171.
doi:<http://dx.doi.org/10.1016/j.energy.2016.10.038>

> Tanikawa, H., & Hashimoto, S. (2009). Urban stock over time: spatial material stock analysis using 4d-GIS. *Building Research & Information*, 37(5-6), 483-502. doi:<https://www.doi.org/10.1080/09613210903169394>

> Treloar, G. J. (1997). Extracting embodied energy paths from input-output tables: towards an input-output-based hybrid energy analysis method. *Economic Systems Research*, 9(4), 375-391.
doi:<https://doi.org/10.1080/09535319700000032>

> Wilson, A., & Boehland, J. (2005). Small is Beautiful U.S. House Size, Resource Use, and the Environment. *Journal of Industrial Ecology*, 9(1-2), 277-287. doi:<https://www.doi.org/10.1162/1088198054084680>

Overall, I think that the authors have done some great work and with some additional nuancing, discussion and more transparency about uncertainty and underestimation of GHG emissions, the paper would be a significant contribution to our climate emergency.

Thank you again for your helpful comments. We are very pleased you think this helps in addressing our climate emergency.

References

- [1] Hertwich EG, *et al.* Material efficiency strategies to reducing greenhouse gas emissions associated with buildings, vehicles, and electronics—a review. *Environmental Research Letters* **14**, 043004 (2019).
- [2] Fishman T, *et al.* Developing scenarios of resource efficiency and climate change: from conception to operation. (2020).
- [3] Pauliuk S, *et al.* Linking Service Provision to Material Cycles—A New Framework for Studying the Resource Efficiency-Climate Change Nexus (RECC). (2020).
- [4] International Energy Agency. CCUS in Clean Energy Transitions. (IEA, Paris, 2020).
- [5] IRP (2020). Resource Efficiency and Climate Change: Material Efficiency Strategies for a Low-Carbon Future. Hertwich, E., Lifset, R., Pauliuk, S., Heeren, N. A report of the International Resource Panel. *United Nations International Resource Panel (IRP): Nairobi, Kenya*.
- [6] Rogelj J, *et al.* Scenarios towards limiting global mean temperature increase below 1.5 °C. *Nature Climate Change* **8**, 325-332 (2018).
- [7] Anderson K, Peters G. The trouble with negative emissions. *Science* **354**, 182-183 (2016).
- [8] van Vuuren DP, *et al.* Alternative pathways to the 1.5 °C target reduce the need for negative emission technologies. *Nature Climate Change* **8**, 391-397 (2018).
- [9] Grubler A, *et al.* A low energy demand scenario for meeting the 1.5 C target and sustainable development goals without negative emission technologies. *Nature energy* **3**, 515-527 (2018).
- [10] Cao Z, *et al.* The sponge effect and carbon emission mitigation potentials of the global cement cycle. *Nature Communications* **11**, 3777 (2020).
- [11] Clark MA, *et al.* Global food system emissions could preclude achieving the 1.5° and 2° C climate change targets. *Science* **370**, 705-708 (2020).
- [12] Pérez-Fortes M, Moya JA, Vatopoulos K, Tzimas E. CO₂ Capture and Utilization in Cement and Iron and Steel Industries. *Energy Procedia* **63**, 6534-6543 (2014).
- [13] Rogelj J, *et al.* Mitigation pathways compatible with 1.5 C in the context of sustainable development. (2018).
- [14] Van Vuuren DP, *et al.* Alternative pathways to the 1.5 C target reduce the need for negative emission technologies. *Nature climate change*

8, 391-397 (2018).

- [15] Hertwich E, Lifset R, Ali S, Pauliuk S, Heeren N, Tu Q. Resource Efficiency and Climate Change: Material Efficiency Strategies for a Low-Carbon Future. Summary for Policy Makers. A report of the International Resource Panel. (Nairobi, Kenya, 2019).
- [16] Shanks W, Dunant C, Drewniok MP, Lupton R, Serrenho A, Allwood JM. How much cement can we do without? Lessons from cement material flows in the UK. *Resources, Conservation and Recycling* **141**, 441-454 (2019).
- [17] Milford RL, Pauliuk S, Allwood JM, Müller DB. The Roles of Energy and Material Efficiency in Meeting Steel Industry CO2 Targets. *Environmental Science & Technology* **47**, 3455-3462 (2013).
- [18] Crawford RH, Bontinck P-A, Stephan A, Wiedmann T, Yu M. Hybrid life cycle inventory methods—A review. *Journal of Cleaner Production* **172**, 1273-1288 (2018).
- [19] Perkins J, Suh S. Uncertainty implications of hybrid approach in LCA: precision versus accuracy. *Environmental science & technology* **53**, 3681-3688 (2019).
- [20] Crawford RH, Stephan A, Prideaux F. A comprehensive database of environmental flow coefficients for construction materials: closing the loop in environmental design. (2019).
- [21] International Energy Agency. 2019 global status report for buildings and construction: Towards a zero-emission, efficient and resilient buildings and construction sector. (2019).
- [22] Steininger KW, Meyer L, Nabernegg S, Kirchengast G. Sectoral carbon budgets as an evaluation framework for the built environment. *Buildings and Cities* **1**, (2020).
- [23] Liu G, Bangs CE, Müller DB. Stock dynamics and emission pathways of the global aluminium cycle. *Nature Climate Change* **3**, 338-342 (2013).
- [24] Creutzig F, *et al.* Towards demand-side solutions for mitigating climate change. *Nature Climate Change* **8**, 260-263 (2018).
- [25] Venkatraj V, Dixit MK, Yan W, Lavy S. Evaluating the impact of operating energy reduction measures on embodied energy. *Energy and Buildings* **226**, 110340 (2020).
- [26] Dixit MK, Fernández-Solís JL, Lavy S, Culp CH. Need for an embodied energy measurement protocol for buildings: A review paper.

Renewable and sustainable energy reviews **16**, 3730-3743 (2012).

- [27] Allwood JM, et al. *Sustainable materials: with both eyes open*. Citeseer (2012).
- [28] Carruth MA, Allwood JM, Moynihan MC. The technical potential for reducing metal requirements through lightweight product design. *Resources, Conservation and Recycling* **57**, 48-60 (2011).
- [29] Bendsoe MP, Sigmund O. *Topology optimization: theory, methods, and applications*. Springer Science & Business Media (2013).
- [30] Ghaffar SH, Corker J, Fan M. Additive manufacturing technology and its implementation in construction as an eco-innovative solution. *Automation in Construction* **93**, 1-11 (2018).
- [31] Dhar S, Pathak M, Shukla PR. Transformation of India's steel and cement industry in a sustainable 1.5 °C world. *Energy Policy*, 111104 (2019).
- [32] Liu G, Xu K, Zhang X, Zhang G. Factors influencing the service lifespan of buildings: An improved hedonic model. *Habitat International* **43**, 274-282 (2014).
- [33] Churkina G, et al. Buildings as a global carbon sink. *Nature Sustainability* **3**, 269-276 (2020).
- [34] Oliver CD, Nassar NT, Lippke BR, McCarter JB. Carbon, Fossil Fuel, and Biodiversity Mitigation With Wood and Forests. *Journal of Sustainable Forestry* **33**, 248-275 (2014).
- [35] Pomponi F, Hart J, Arehart JH, D'Amico B. Buildings as a Global Carbon Sink? A Reality Check on Feasibility Limits. *One Earth* **3**, 157-161 (2020).
- [36] Buchanan MK, Kulp S, Cushing L, Morello-Frosch R, Nedwick T, Strauss B. Sea level rise and coastal flooding threaten affordable housing. *Environmental Research Letters* **15**, 124020 (2020).
- [37] Xu C, Kohler TA, Lenton TM, Svenning J-C, Scheffer M. Future of the human climate niche. *Proceedings of the National Academy of Sciences* **117**, 11350 (2020).
- [38] Pauliuk S, Milford RL, Müller DB, Allwood JM. The steel scrap age. *Environmental science & technology* **47**, 3448-3454 (2013).
- [39] Stehfest E, van Vuuren D, Bouwman L, Kram T. *Integrated assessment of global environmental change with IMAGE 3.0: Model description and policy applications*. Netherlands Environmental Assessment Agency (PBL) (2014).

- [40] Pauliuk S, Arvesen A, Stadler K, Hertwich EG. Industrial ecology in integrated assessment models. *Nature Climate Change* **7**, 13-20 (2017).
- [41] Leclère D, *et al.* Bending the curve of terrestrial biodiversity needs an integrated strategy. *Nature* **585**, 551-556 (2020).
- [42] Luderer G, *et al.* Residual fossil CO₂ emissions in 1.5–2 C pathways. *Nature Climate Change* **8**, 626-633 (2018).
- [43] Mendoza Beltran A, *et al.* When the background matters: using scenarios from integrated assessment models in prospective life cycle assessment. *Journal of Industrial Ecology* **24**, 64-79 (2020).
- [44] Miatto A, Schandl H, Tanikawa H. How important are realistic building lifespan assumptions for material stock and demolition waste accounts? *Resources, Conservation and Recycling* **122**, 143-154 (2017).
- [45] Fishman T, Schandl H, Tanikawa H, Walker P, Krausmann F. Accounting for the material stock of nations. *Journal of Industrial Ecology* **18**, 407-420 (2014).
- [46] Tanikawa H, Fishman T, Okuoka K, Sugimoto K. The weight of society over time and space: A comprehensive account of the construction material stock of Japan, 1945–2010. *Journal of Industrial Ecology* **19**, 778-791 (2015).
- [47] Kapur A, Keoleian G, Kendall A, Kesler SE. Dynamic modeling of in-use cement stocks in the United States. *Journal of Industrial Ecology* **12**, 539-556 (2008).
- [48] Deetman S, Marinova S, van der Voet E, van Vuuren DP, Edelenbosch O, Heijungs R. Modelling global material stocks and flows for residential and service sector buildings towards 2050. *Journal of Cleaner Production*, 118658 (2019).
- [49] Arehart JH, Hart J, Pomponi F, D'Amico B. Carbon sequestration and storage in the built environment. *Sustainable Production and Consumption* **27**, 1047-1063 (2021).
- [50] Krausmann F, *et al.* Global socioeconomic material stocks rise 23-fold over the 20th century and require half of annual resource use. *Proceedings of the National Academy of Sciences* **114**, 1880-1885 (2017).
- [51] Tanikawa H, Hashimoto S. Urban stock over time: spatial material stock analysis using 4d-GIS. *Building Research & Information* **37**, 483-502 (2009).
- [52] Stephan A, Athanassiadis A. Towards a more circular construction sector: Estimating and spatialising current and future non-structural material replacement flows to maintain urban building stocks. *Resources, Conservation and Recycling* **129**, 248-262 (2018).

- [53] Wilson A, Boehland J. Small is beautiful US house size, resource use, and the environment. *Journal of Industrial Ecology* **9**, 277-287 (2005).
- [54] Stephan A, Crawford RH. The relationship between house size and life cycle energy demand: Implications for energy efficiency regulations for buildings. *Energy* **116**, 1158-1171 (2016).

REVIEWER COMMENTS

Reviewer #1 (Remarks to the Author):

Thank you for seeking to carefully address all issues I raised, and for a referenced and detailed argumentation why and how you did. While most of these changes are very fine, I do have three issues and suggestions below.

Thanks also for your recalculations. However, there seem to be a core scaling error either in calculation or in result representation. For the interpretation of the emission reduction potential: The respective data in the supplementary xls file (in Mt CO₂eq), and correctly specified in the Figure 2, is repeatedly wrongly translated and mentioned in the main text. E.g. for the M1-scenario (more intensive use) Figure 2 gives a reduction potential of 5922.61 Mt CO₂eq globally, but in the text is mentioned as corresponding to 59 Gt CO₂eq – but actually is only 5.9 Gt CO₂eq. The same holds for all other reduction potentials (M2-M7). I thus wonder how the overall reduction potential can be in the order of magnitude of 80 GtCO₂eq (as the article continues), when the sum of reduction sub-potentials (even though the pure sum including some double counting) is an order of magnitude (factor 10) smaller.

Let me come back to my earlier issues, though, as well. In continuing our dialogue I do have the following remaining issues which I suggest the authors to take care of, and once the above serious data issue is solved, I hope then would recommend publication.

(1) Lightweight design for concrete: The addition of this new scenario (Table 1) is warmly acknowledged. While I understand and agree to your position of being at the very cautious side in your quantifications, my suggestion is to mention the technological potential already seen (e.g. Hansemann et al., 2021), meeting all building standard regulations at least as well as currently used practices. In other words, the argument for your current quantified scenario seems appropriate, yet my suggestion is to indicate the substantially larger range already practiced at the object scale (with the barrier being the ~10% higher cost to date). After the sentence in line 200-204 would be one option to place this information.

(2) New technologies of CCU in cement production to be relevant at significant scale only “long term” (line 268): Similarly, I consider your argumentation, that you consider only technologies that are commercially available and widely documented in recent literature as relevant for your quantification. However, with the world leading companies e.g. in cement production having their plans of the table (and pilots ahead) of all production switching to full CCU well before 2050, I consider it misleading to indicate that this is relevant only in the “long term” (when “long term” in your article refers to beyond 2060). My suggestion is to make it fully transparent that your technologies used for quantification are based on technologies “that are commercially available and widely documented in recent literature”, but that substantial further developments could take place also up to 2060.

(3) As wood is mentioned as alternative building material with better environmental performance (line 58, and more extensively, also referenced, line 351), my recommendation is to inform about the complexity of such correct accounting to indicate that the use of wood is not per se connected to large GHG concentration improvements. The Methods section might be a place to point his out.

What I am concerned with respect to emissions accounting of biomass materials and products is the following:

Domestic forests serve as a natural CO₂ sink. As such, forests are indispensable to curb greenhouse gas concentrations in our atmosphere. This atmospheric-concentration-reducing effect needs to be preserved when biomass is used more intensively across the energy, building and industry sectors.

In this respect, when substituting conventional materials by biomass one has to be careful when accounting for the impacts of activities of the building and industry sector on the greenhouse gas concentration in the atmosphere. The reason is, that not the wood-based (construction) material itself has a CO₂ fixing effect but the forest from where the wood came from. This means, that the CO₂-fixing effect from the forest is transferred to the building or industry sector. Hence, when accounting for the emission-fixing effect of wood-based products, various parameters such as the origin of the wood (e.g. domestic, sustainable forest management) and rotation periods of harvesting are crucial. In terms of emissions accounting of wood-based products, when looking, for example, at the building sector, static approaches have long been present. Static approaches typically adopted either ignore the CO₂ uptake or account for it by estimating the carbon content of the wood and deriving the amount of CO₂ "fixed". Recent research advocates towards a dynamic modelling, where the temporal effect of a slow, gradual uptake in the forest is taken into account, along with other important aspects such as tree rotation period, see e.g. Hoxha et. al. (2020). The static approach is found to underestimate the actual contribution of (biomass-based) building components to increase atmospheric greenhouse gas concentrations ((Hoxha et al., 2020) use a global warming score) by at least 35% and up to 200%, with the single most relevant parameter governing the deviation relying on the rotation period of the forest, which is not considered in static approaches at all as the system boundary is limited to the product stage. When restricting the analysis to the product stage, most often even a negative emission contribution (a "sink") is indicated, even though the actual overall impact of these building components based on wood may be clearly not negative, but connected to positive net emissions.

Furthermore, when looking at a national GHG balance, one has to be careful to not double-count the emission fixing effect of wood. The reason is that the effect of forests as a CO₂ sink is usually assigned to the emission sector "Land-Use, Land-Use Change and Forestry" (LULUCF). Hence, when presenting the emission-fixing effect of wood-based materials in the building and industry sectors, one must clarify whether those emissions already have been accounted for in the LULUCF or not, and accommodate results (and time periods they refer to) accordingly.

As this is not a core issue of your paper, but you mention (actually once carefully) this biomass/wood option, the readers should have access to this crucial additional information.

References:

Hansemann, G., Schmid, R., Holzinger, C., Tapley, J. P., Peters, S., Trummer, A. & Kupelwieser, H., 2021, Lightweight Reinforced Concrete Slab: 130 different 3D printed voids, In : CPT Worldwide - Construction Printing Technology. June 2021, 2, p. 68 73 p.

Hoxha, E., Passer, A., Saade, M.R.M., Trigaux, D., Shuttleworth, A., Pittau, F., Allacker, K., Habert, G., 2020. Biogenic carbon in buildings: a critical overview of LCA methods. *Build. Cities* 1, 504–524. <https://doi.org/10.5334/bc.46>

Reviewer #2 (Remarks to the Author):

Comments have been addressed. I recommend this paper for publication.

Reviewer #3 (Remarks to the Author):

The authors have done a good job at addressing my previous comments and have significantly improved the manuscript.

For me, the manuscript can be accepted for publication.

I would just kindly ask the authors to fix the following minor issues before submitting the final manuscript.

1- References: Please include the digital object identified for all journal papers (and where relevant) and unbold the issue of the journal papers.

2- References: ref 99, please use the correct citation to the book reference: Crawford, R. H., Stephan, A., & Prideaux, F. (2019). Environmental Performance in Construction (EPiC) database. Melbourne: The University of Melbourne.

Best wishes and have a good summer.
Prof André Stephan

Detailed responses to reviewers' comments

Thank you for your email and to all reviewers for their valuable comments on our manuscript. We have carried out a comprehensive revision of the manuscript in accordance with their comments and our detailed responses to reviewers' comments are listed as follows:

R1 comments

Reviewer's comments	Response
Thank you for seeking to carefully address all issues I raised, and for a referenced and detailed argumentation why and how you did. While most of these changes are very fine, I do have three issues and suggestions below.	Thank you for your comments, we appreciate the constructive comments and spotting the typo in the number below. Please find below the detailed responses.
Thanks also for your recalculations. However, there seem to be a core scaling error either in calculation or in result representation. For the interpretation of the emission reduction potential: The respective data in the supplementary xls file (in Mt CO₂eq), and correctly specified in the Figure 2, is repeatedly wrongly translated and mentioned in the main text. E.g. for the M1-scenario (more intensive use) Figure 2 gives a reduction potential of 5922.61 Mt CO₂eq globally, but in the text is mentioned as corresponding to 59 Gt CO₂eq – but actually is only 5.9 Gt CO₂eq. The same holds for all other reduction potentials (M2-M7). I thus wonder how the overall reduction potential can be in the order of magnitude of 80 GtCO₂eq (as the article continues), when the sum of reduction	Thank you for catching this. This was actually a typo in scaling term in the code for Figure 2, and we have updated accordingly. The figure as a whole is updated. For the example you mention, the M1- more intensive use yields a reduction of 59226.1 Mt (59.2 Gt) CO₂eq, instead of 5922.61 Mt (5.92 Gt) CO₂eq. We have double-checked throughout the paper and code. [see supplementary Source Data and lines 161-163 in the main text]

sub-potentials (even though the pure sum including some double counting) is an order of magnitude (factor 10) smaller.	
Let me come back to my earlier issues, though, as well. In continuing our dialogue I do have the following remaining issues which I suggest the authors to take care of, and once the above serious data issue is solved, I hope then would recommend publication. (1) Lightweight design for concrete: The addition of this new scenario (Table 1) is warmly acknowledged. While I understand and agree to your position of being at the very cautious side in your quantifications, my suggestion is to mention the technological potential already seen (e.g. Hansemann et al., 2021), meeting all building standard regulations at least as well as currently used practices. In other words, the argument for your current quantified scenario seems appropriate, yet my suggestion is to indicate the substantially larger range already practiced at the object scale (with the barrier being the ~10% higher cost to date). After the sentence in line 200-204 would be one option to place this information.	Thank you for your publication recommendation. Please see below for a detailed response to your suggestions. For concrete light weighting, following your suggestion, we have now added this information after the sentence in lines 200-204. The additional text now reads: Some adjustment of building regulations is likely essential for such light-weighting transitions. Depending on the technologies and level of adoption there may be larger opportunities for light-weighting than those adopted in Table 1, e.g., 20% or more concrete reduction (Hansemann et al., 2021; IRP, 2020)^{1, 2}. The current cost barriers to this implementation may reduce over time through deployment-led learning. [see lines 195-199 in the main text]
(2) New technologies of CCU in cement production to be relevant at significant scale only “long term” (line 268): Similarly, I consider your argumentation, that you consider only technologies that are commercially available and widely documented in recent literature as relevant for your quantification. However, with the world leading companies e.g. in cement	Thank you for this suggestion. We have now rephrased the text and the updated sentences now read: These technologies, and Negative Emission Technologies (NETs) which remove carbon emissions directly from the atmosphere, are still in early development and

production having their plans of the table (and pilots ahead) of all production switching to full CCU well before 2050, I consider it misleading to indicate that this is relevant only in the “long term” (when “long term” in your article refers to beyond 2060). My suggestion is to make it fully transparent that your technologies used for quantification are based on technologies “that are commercially available and widely documented in recent literature”, but that substantial further developments could take place also up to 2060.	face significant technological and socioeconomic barriers^{3, 4}. Although substantial further developments could take place up to 2060, we consider them as a complement to existing and more predictable technologies (e.g. recycling) and regulatory developments (e.g. building longevity), as broadly highlighted in the literature^{2, 5}. [see lines 259-265 in the main text]
(3) As wood is mentioned as alternative building material with better environmental performance (line 58, and more extensively, also referenced, line 351), my recommendation is to inform about the complexity of such correct accounting to indicate that the use of wood is not per se connected to large GHG concentration improvements. The Methods section might be a place to point his out. What I am concerned with respect to emissions accounting of biomass materials and products is the following: Domestic forests serve as a natural CO₂ sink. As such, forests are indispensable to curb greenhouse gas concentrations in our atmosphere. This atmospheric-concentration-reducing effect needs to be preserved when biomass is used more intensively across the energy, building and	Thank you for this very interesting point and for suggesting Hoxha et al., 2020. Following your suggestion, we have now added a note on this in the Method section. The updated text now reads: Another improvement of the LCI database could include accounting for the carbon sequestration effect of wood-based products using dynamic sub-models to capture the temporal effect of a slow, gradual uptake of carbon in forests, along with other important factors such as the origin and rotation periods of harvesting (Hoxha et al., 2020)⁶. A similar improvement could also include a dynamic sub-model to incorporate CO₂ reabsorption for concrete once construction is complete⁷. [see lines 483-487 in the main text]

industry sectors.

In this respect, when substituting conventional materials by biomass one has to be careful when accounting for the impacts of activities of the building and industry sector on the greenhouse gas concentration in the atmosphere. The reason is, that not the wood-based (construction) material itself has a CO₂ fixing effect but the forest from where the wood came from. This means, that the CO₂-fixing effect from the forest is transferred to the building or industry sector. Hence, when accounting for the emission-fixing effect of wood-based products, various parameters such as the origin of the wood (e.g. domestic, sustainable forest management) and rotation periods of harvesting are crucial. In terms of emissions accounting of wood-based products, when looking, for example, at the building sector, static approaches have long been present. Static approaches typically adopted either ignore the CO₂ uptake or account for it by estimating the carbon content of the wood and deriving the amount of CO₂ “fixed”. Recent research advocates towards a dynamic modelling, where the temporal effect of a slow, gradual uptake in the forest is taken into account, along with other important aspects such as tree rotation period, see e.g. Hoxha et. al. (2020). The static approach is found to underestimate the actual contribution of (biomass-based) building components to increase atmospheric greenhouse gas concentrations ((Hoxha et al., 2020) use a global warming score) by at

least 35% and up to 200%, with the single most relevant parameter governing the deviation relying on the rotation period of the forest, which is not considered in static approaches at all as the system boundary is limited to the product stage. When restricting the analysis to the product stage, most often even a negative emission contribution (a “sink”) is indicated, even though the actual overall impact of these building components based on wood may be clearly not negative, but connected to positive net emissions.

Furthermore, when looking at a national GHG balance, one has to be careful to not double-count the emission fixing effect of wood. The reason is that the effect of forests as a CO₂ sink is usually assigned to the emission sector “Land-Use, Land-Use Change and Forestry” (LULUCF). Hence, when presenting the emission-fixing effect of wood-based materials in the building and industry sectors, one must clarify whether those emissions already have been accounted for in the LULUCF or not, and accommodate results (and time periods they refer to) accordingly.

As this is not a core issue of your paper, but you mention (actually once carefully) this biomass/wood option, the readers should have access to this crucial additional information.

References:

Hansemann, G., Schmid, R., Holzinger, C., Tapley, J. P., Peters, S., Trummer, A. & Kupelwieser, H., 2021, Lightweight Reinforced Concrete Slab: 130 different 3D printed voids, In : CPT Worldwide - Construction Printing Technology. June 2021, 2, p. 68 73 p. Hoxha, E., Passer, A., Saade, M.R.M., Trigaux, D., Shuttleworth, A., Pittau, F., Allacker, K., Habert, G., 2020. Biogenic carbon in buildings: a critical overview of LCA methods. Build. Cities 1, 504–524. https://doi.org/10.5334/bc.46	
---	--

R2 comments

Reviewer's comments	Response
Comments have been addressed. I recommend this paper for publication.	Thank you again for all comments and for recommending this paper for publication.

R3 comments

Reviewer's comments	Response
The authors have done a good job at addressing my previous comments and have significantly improved the manuscript.	Thank you again for your comments and for recommending this paper for publication.

For me, the manuscript can be accepted for publication.	
I would just kindly ask the authors to fix the following minor issues before submitting the final manuscript. 1- References: Please include the digital object identified for all journal papers (and where relevant) and unbold the issue of the journal papers. 2- References: ref 99, please use the correct citation to the book reference: Crawford, R. H., Stephan, A., & Prideaux, F. (2019). Environmental Performance in Construction (EPiC) database. Melbourne: The University of Melbourne.	Thank you for pushing us to address reference-related issues. 1- References: We have included the DOI and made the issue numbers non-bold for papers where applicable. 2- References: We have now updated the book reference (ref 98) as you suggested.

References

- [1] Hansemann G, *et al.* Lightweight Reinforced Concrete Slab: 130 different 3D printed voids. *CPT Worldwide-Construction Printing Technology* 2021, 68 (2021).
- [2] IRP. Resource Efficiency and Climate Change: Material Efficiency Strategies for a Low-Carbon Future. Hertwich, E., Lifset, R., Pauliuk, S., Heeren, N. A report of the International Resource Panel. *United Nations International Resource Panel (IRP): Nairobi, Kenya*, (2020).
- [3] IRP (2020). Resource Efficiency and Climate Change: Material Efficiency Strategies for a Low-Carbon Future. Hertwich, E., Lifset, R., Pauliuk, S., Heeren, N. A report of the International Resource Panel. *United Nations International Resource Panel (IRP): Nairobi, Kenya*.
- [4] Anderson K, Peters G. The trouble with negative emissions. *Science* 354, 182-183 (2016).
- [5] Hertwich E, Lifset R, Ali S, Pauliuk S, Heeren N, Tu Q. Resource Efficiency and Climate Change: Material Efficiency Strategies for a Low-Carbon Future. Summary for Policy Makers. A report of the International Resource Panel. (Nairobi, Kenya, 2019).
- [6] Hoxha E, *et al.* Biogenic carbon in buildings: a critical overview of LCA methods. *Buildings and Cities*, (2020).
- [7] Cao Z, *et al.* The sponge effect and carbon emission mitigation potentials of the global cement cycle. *Nature Communications* 11, 3777 (2020).

REVIEWERS' COMMENTS

Reviewer #1 (Remarks to the Author):

The data scaling error in Figure 2 has been corrected, and my remaining three suggestions have been well addressed by the authors. I therefore clearly recommend this paper for publication.

Detailed responses to reviewers' comments

Thank you for your email and to the reviewer for the valuable comments on our manuscript. See our detailed responses to reviewers' comments are listed as follows:

R1 comments

Reviewer's comments	Response
The data scaling error in Figure 2 has been corrected, and my remaining three suggestions have been well addressed by the authors. I therefore clearly recommend this paper for publication.	Thank you again for catching this, for the fruitful discussion, and your suggestions.